# Electron cryo-microscopy reveals the structure of the archaeal thread filament

Matthew C. Gaines[1,2], Michail N. Isupov[3], Shamphavi Sivabalasarma[4,5], Risat Ul Haque[1,2], Mathew McLaren[1,2], Clara L. Mollat[4], Patrick Tripp[4], Alexander Neuhaus[1,2], Vicki A. M. Gold [1,2], Sonja-Verena Albers [4,5,6] & Bertram Daum [1,2]✉

Pili are filamentous surface extensions that play roles in bacterial and archaeal cellular processes such as adhesion, biofilm formation, motility, cell-cell communication, DNA uptake and horizontal gene transfer. The model archaeaon *Sulfolobus acidocaldarius* assembles three filaments of the type-IV pilus superfamily (archaella, archaeal adhesion pili and UV-inducible pili), as well as a so-far uncharacterised fourth filament, named "thread". Here, we report on the cryo-EM structure of the archaeal thread. The filament is highly glycosylated and consists of subunits of the protein Saci_0406, arranged in a head-to-tail manner. Saci_0406 displays structural similarity, but low sequence homology, to bacterial type-I pilins. Thread subunits are interconnected via donor strand complementation, a feature reminiscent of bacterial chaperone-usher pili. However, despite these similarities in overall architecture, archaeal threads appear to have evolved independently and are likely assembled by a distinct mechanism.

The ability to adhere to surfaces and form biofilms is essential to microbial life and the development of disease. Surface adhesion and biofilm formation are promoted by various filaments that extend from the prokaryotic cell surface[1–4]. In bacteria, a large variety of adhesive filaments have been well studied. These include chaperone-usher (CU), curli, type-IV (T4P) and type-V pili (T5P) in Gram-negative bacteria, and sortase-dependent pili in Gram-positive bacteria[5–11].

In archaea, cell adhesion and biofilm formation are far less understood. So far, the best studied archaeal adhesive filaments are of the type-IV pilus superfamily (T4FF)[12–15]. T4FF are conserved in bacteria and archaea and consist of helically organised lollipop-shaped subunits, called type-IV pilins (T4P). T4P are hallmarked by a conserved hydrophobic N-terminal α-helix and a variable globular C-terminus rich in β-strands. T4FF have a common assembly pathway. The subunits are initially expressed as pre-pilins and inserted into the cell

membrane by the Sec translocon[16]. Upon cleavage of the N-terminal signal peptide by a Class-III signal peptidase, the pilins are primed for the insertion into a nascent T4FF filament in a process that is catalysed by a membrane-spanning and ATPase-driven multiprotein complex[17]. In the assembled pilus, the α-helices form the core of the filament, while the globular domains face the extracellular medium[18]. Co-option and adaptation of homologous genes in the T4FF have resulted in a variety of T4P-like filaments with additional functions such as motility and DNA uptake[19,20].

Recent evidence suggests the existence of another type of archaeal adhesive filament[21], which shares characteristics reminiscent of bacterial chaperone-usher (CU) pili. These are surface filaments, that play a key role in biofilm formation and virulence of pathogenic strains[22]. CU pili include Type I and P pili, which are found in uro-pathogenic *Escherichia coli*, and enable those bacteria to adhere to

[1]Living Systems Institute, University of Exeter, Stocker Road, EX4 4QD, Exeter, UK. [2]Department of Biosciences, Faculty of Health and Life Sciences, Stocker Road, EX4 4QD, Exeter, UK. [3]Henry Wellcome Building for Biocatalysis, Department of Biosciences, Faculty of Health and Life Sciences, University of Exeter, EX4 4QD, Exeter, UK. [4]Institute of Biology II, Molecular Biology of Archaea, University of Freiburg, Schänzlestraße 1, 79104 Freiburg, Germany. [5]Spemann Graduate School of Biology and Medicine, University of Freiburg, Freiburg, Germany. [6]Signalling Research Centres BIOSS and CIBBS, Faculty of Biology, University of Freiburg, Freiburg, Germany. ✉e-mail: b.daum2@exeter.ac.uk

endothelial cell surfaces[22,23]. Cell-surface attachment is mediated by specialised adhesins that cap the filaments' distal termini and are selective to sugar residues in glycolipid receptors on the host's cell surfaces[24,25].

Fimbriae are hollow fibres that are assembled from pilin like subunits[11]. These subunits are initially expressed as precursor proteins and secreted by the Sec translocon into the periplasmic space[26]. Subunits for these filaments have a conserved structure consisting of an N-terminal (β-strand) extension (NTE) and a C-terminal globular domain with an incomplete immunoglobin-like fold (head). In the assembled filament, each subunit inserts its tail into the β-barrel of an adjacent subunit through a process called donor strand complementation (DSC)[27]. This mechanism, which entails the formation of several hydrogen bonds, concatenates the subunits into a highly stable filament[28].

Assembly into a filament takes place in the periplasm and is mediated by the chaperone-usher (CU) pathway[29]. Here, the chaperone (FimC in Type1 and PapD in P-pili of *E. coli*) binds to the nascent N-terminus of a newly synthesised pilin and aids its integration into the growing pilus[30]. An outer membrane integral barrel protein called usher guides the fimbriae through the outer membrane, from which they finally extend up to 3000 subunits into the surrounding medium[31–33]. Assembly via DSC is also found in type-V pili (T5P) in the Gram- negative bacteria *Porphyromonas gingivalis*[34] and *Salmonella enterica*[35,36], as well as in sortase-dependent pili (SDP) in the Gram-positive bacteria *Actinomyces naeslundii*, *Corynebacterium diphtheriae*, *Enterococcus faecalis*, and various *Streptococcus* species[37]. However, T5P and SDP are assembled via mechanisms distinct from the CU pathway[38].

The crenarchaeon *Sulfolobus acidocaldarius* serves as an intriguing model organism to study cell surface structures, as it thrives at 75 °C and pH 3 and hence surface structures are adapted to these extreme environmental conditions[39]. *S. acidocaldarius* assembles three surface filaments of the T4FF family; an archaellum, archaeal adhesion pili (Aap), UV inducible pili (Ups) that facilitate DNA exchange under UV stress, as well as a novel and a so-far uncharacterised filament, named "thread"[40].

When Aap, UV pili and the archaella are deleted in *S. acidocaldarius*, the cells still form threads[40]. These are thin, 5 nm wide filaments that are highly abundant on the cell surface reaching up to several microns in length. Here, we present a 3.45 Å resolution structure of this filament and identify its subunit protein. The subunits are tightly interlinked via DSC, akin bacterial CU-assembled pili. In addition, the subunits appear to be covalently linked via unusual N-terminal isopeptide bonds. We find that the threads are highly glycosylated and based on our cryoEM map, we were able to model five complete N-glycan structures per subunit into our structure. Structure and sequence-based bioinformatic analyses reveal a hitherto unknown archaea-specific class of cell surface filament.

## Results
### Thread assembly is independent of Signal Peptide III
Although threads were already observed in the early 2000s[40] in *S. acidocaldarius* cells, their structure and assembly mechanism has been elusive. The prepilin signal peptidase PibD (or class III signal peptide peptidase SPIII) is crucial for the assembly and function of archaella, UV-pili and Aap pili[41]. We therefore asked whether it is also essential for the assembly of the threads. A *ΔpibD* deletion mutant showed thread filaments on its surface as analysed by Transmission electron microscopy (TEM) (Supplementary Fig. 1a–c). Hence, threads are formed by a PibD independent pathway and thus do not belong to the T4FF superfamily.

### CryoEM and helical reconstruction of thread filaments
We aimed to elucidate the structure of the thread filament by electron cryo-microscopy (cryo-EM). To this end, an *S. acidocaldarius* strain

expressing only AAP and threads and no archaella or UV pili (MW158) was grown to stationary phase. This strain was chosen for two reasons. Firstly, it allowed us to co-purify the threads alongside a reference filament with established purification parameters and genetic background. Secondly, analysing two structurally distinct filaments in one sample would allow us to compare their structures from the same micrographs in future studies.

The filaments were sheared from the cells and further purified by CsCl gradient centrifugation. The resulting sample containing a mixed population of threads and Aap pili was plunge frozen on cryoEM grids, from which 6272 cryoEM movies were recorded (Supplementary Table 1). Contrast transfer function (CTF) correction and motion correction was performed using CryoSPARC[42], before filaments were picked automatically using the filament tracer programme. The dataset was cleaned up by several rounds of 2D classification to remove erroneously autopicked Aap pili and poorly resolved classes of threads (Supplementary Fig. 2). From our 2D classifications, it became apparent that threads often align into cables consisting of several parallel strands of filaments (Supplementary Fig. 1h). Similar cables were observed in micrographs of negatively stained *S. acidocaldarius* cells (Supplementary Fig. 1c), indicating that the threads have the tendency to adhere to each other.

The Helix Refine routine within cryoSPARC allowed for low resolution 3D reconstructions of filaments without the need to impose helical parameters[42]. The resulting map with a resolution of 5.54 Å clearly showed that the thread filament consists of a helical string of subunits (Supplementary Fig. 3) and allowed us to determine approximate helical parameters in Real Space. A symmetry search around these values in cryoSPARC resulted in a clear peak at a rise of 31.6 Å, which was corroborated by a meridional reflection in Fourier power spectra calculated with SPRING, and a twist of −103.2° (Supplementary Fig. 4)[43].

Applying these values to the 3D refinement resulted in a map of 4.02 Å resolution, which further improved to 3.45 Å after several rounds of CTF refinement. A local resolution estimation showed that the map was best resolved in the core of the filament where it reached 3.0 Å resolution (Supplementary Fig. 5). The final map revealed that the thread filament consists of globular protein subunits that are arranged in a head-to-tail manner (Fig. 1a).

### Identifying the thread subunit protein
We sought to identify the thread subunit via mass spectrometry, however the filament was resistant to digestion with trypsin or treatment with guanidine hydrochloride (GdnHCl). This remarkable stability was also observed for the LAL/Aap pilus of *Sulfolobus islandicus*, as well as for *Sulfolobus solfataricus*[12,44].

Due to the quality of our map, in particular the well resolved glycosylation sites, the subunit of the thread was identified from its structure alone. This was done without need for the gene sequence, similar to previously described strategies (Supplementary Fig. 6)[44–46]. We initially built a poly-alanine chain into our map to determine the direction and length of the backbone. This revealed that the polypeptide sequence is approximately 206 amino acids long. We were able to pinpoint the position of characteristic amino acids along the backbone due to their distinctive shape and size. These included aromatic amino acids, as well as prolines. In addition, we recognised five glycosylation sites, which appear typically as distinct densities that protrude from the backbone and are too large to correspond to amino acid side chains. In *S. acidocaldarius*, N-linked glycosylation has been investigated biochemically and via mass spectrometry[47]. The glycans occur at conserved consensus sites with the sequence NXT/S and usually consist of a tri-branched hexasaccharide containing the unusual sugar 6-sulfoquinovose[47]. Using the position of these glycans, as well as those of our modelled characteristic amino acids as a fingerprint, we blasted against the proteome of *S. acidocaldarius* for

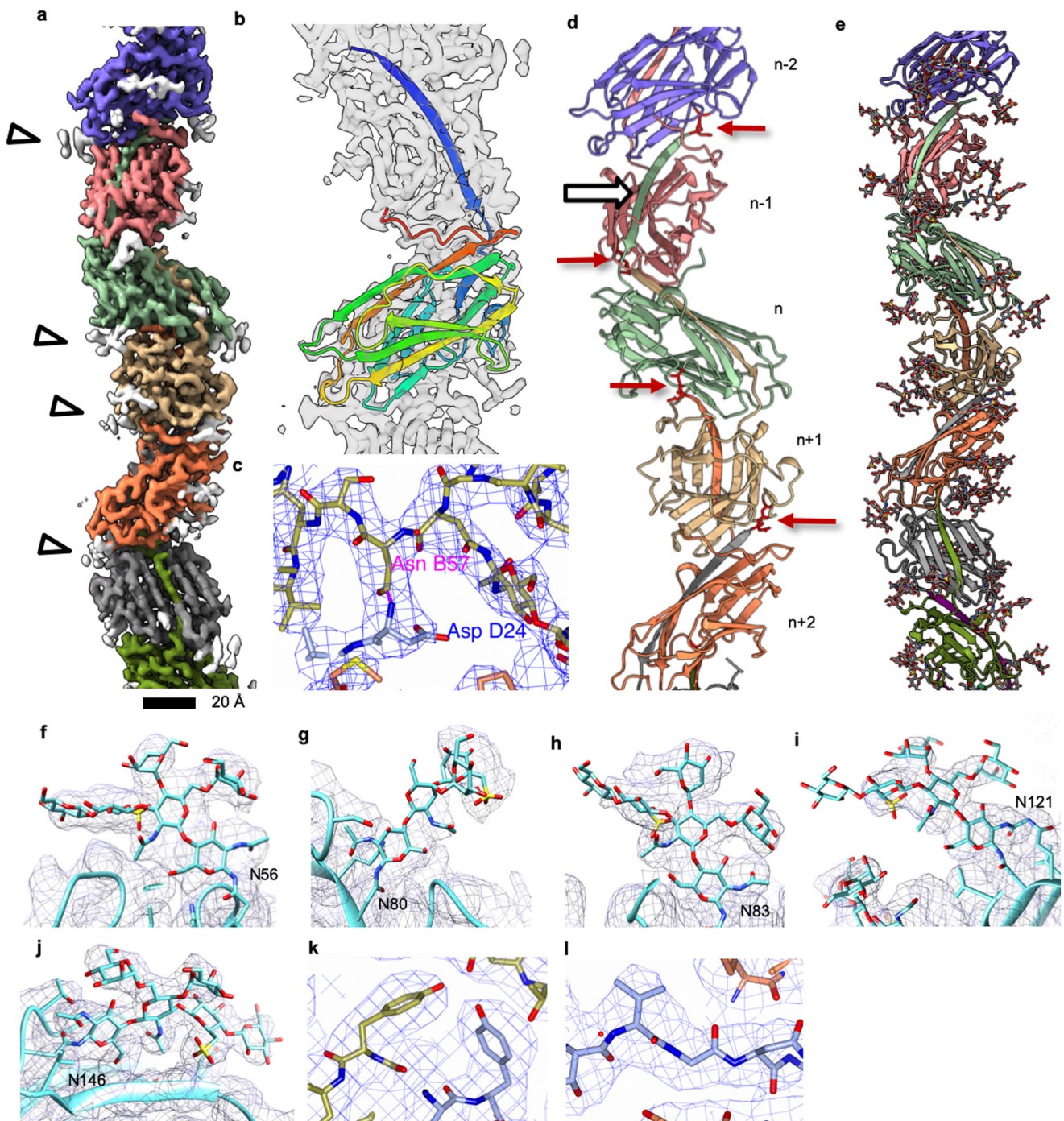

**Fig. 1 | Helical reconstruction and atomic model of the thread. a** Segmented surface representation of the cryoEM map showing each subunit in a different colour. Glycan densities are shown in white (white arrowheads). **b** The atomic model of one Saci_0406 subunit (rainbow) fitted into the cryoEM map (transparent grey) N-terminus, blue; C-terminus, red. Glycans not shown for simplicity. **c** Intermolecular isopeptide bond is shown as a dashed purple line. The N-terminal residue Asp24 of a subunit ($n + 2$; carbons in ice blue) appears to be covalently bound to Asn57 two subunits along the chain (n; carbons in gold). **d** Atomic model of 5 consecutive Saci_0406 subunits in ribbon representation. Glycans are not shown for simplicity. A white arrow indicates the location of the donor strand complementation between the N-terminal tail domain of subunit (*n*) and the C-terminal head domain of the previous subunit in the chain (*n* − 1). Red arrows indicate the location of isopeptide bonds. The tail domain of each subunit n is partially buried in the two consecutive subunits *n* − 1 and *n* − 2. **e** Atomic model of the thread filament with glycans in stick representation. **f–j** Closeups of the five glycosylation sites found in Saci_0406. Atomic models of the glycans are in stick, the polypeptide backbone in ribbon representation. The CryoEM density map is shown as grey mesh. **k, l** Closeups of map and fitted atomic model demonstrating the quality of the data. The colour scheme is as in C, chain (n-1) shows carbons in coral. Scale bar 20 Å.

candidate proteins. This resulted in a single hit, Saci_0406 (Supplementary Fig. 7).

To confirm that Saci_0406 comprises the thread, we attempted to knock out the *saci_0406* gene. The deletion mutant Δ*saci_0406* did not yield any colonies. However, we were able to produce a double deletion Δ*saci_0405/0406* strain. Saci_0405 is the gene next to Saci_0406 and predicted to be transcribed along with it. The fact the Δ*saci_0406* mutant does not yield colonies, but the Δ*saci_0405/0406* does, suggests that expressing Saci_0405 alone may be toxic to the cells. Negative stain electron microscopy showed that the mutant

could assemble Aap pili but not threads, confirming our structural data (Supplementary Fig. 1f).

Using the sequence of Saci_0406, we were able to build an atomic model for the thread unambiguously (Fig. 1b–e), excluding the N-terminus (Met1 – Ala23). We then predicted the structure of this protein using Alphafold2. The resulting prediction closely matched our ab initio structure (Supplementary Fig. 8), indicating that Saci_0406 indeed forms the subunit of the thread filaments. The Alphafold model suggested that Met1 – Ala23 of Saci_0406 form an α-helical signal peptide, which was confirmed by the DeepTMHMM[48] server (Supplementary Fig. 9a, b). Analysing the sequence of Saci_0406 with the SignalP5 server predicted a signal peptidase I processing site, cleaving after the N-terminal α-helix between the position Ala23-Asp24 (AA sequence: ...LVA/DV...) (Supplementary Fig. 9d)[49] and explaining the absence of Met1 – Ala23 in our structure.

## Structure of the thread

Our atomic model reveals that each subunit consists of an N-terminal β-strand (tail) followed by a globular (head) domain. The head domain contains two β-sheets composed of 11 β-strands (Fig. 1b). On the subunit level, one β-strand is missing to complete the sheet, leaving a gap between β-4 and β-12 (Supplementary Fig. 10). Upon closer investigation of the assembled filament, this missing strand is replaced by the β1 tail strand of the next subunit along the filament (n+1) (Fig. 1D; Supplementary Fig. 10b). Each head domain hereby forms a mainly antiparallel beta-blade of a mixed type, comprising 12 beta-strands in total (Supplementary Fig. 10). Such DSC has previously been observed in bacterial CU pili (T1P, P-pili), as well as T5P, where it has been proposed to contribute to the stability of the filament (Supplementary Fig. 11)[35,50]. In the thread, the β-tail from subunit n, extends further into the subunit two positions along the filament (n − 2; Fig. 1d, Supplementary Fig. 10b). β;− 1 of the subunit n is partially buried in the subunits n − 1 and n − 2 (Fig. 1d, Supplementary Fig. 16i), further adding to the stability of the interaction. Moreover, a strong continuous (backbone-like) density connects the carboxamide side chain belonging to Asn57 of each subunit (n) with the α-amino group of the N-terminal Asp24 in the protomer two positions along the filament (n + 2). (Fig. 1c, Supplementary Fig. 12b, Supplementary movie 1). This continuous density, as well as the distance and orientation of the involved residues are typical for isopeptide bonds that are found in SDP pili of Gram-positive bacteria (Supplementary Fig. 12a)[51–54]. A comparison was conducted between the isopeptide bonds of Spy0128 (Streptococcus pyogenes) and Saci_0406. In the bacterial species, the joining of Threonine 311 to Lysine 161 in the gram-positive pili releases a water molecule, whereas in the case of the thread an unusual ammonium ion is produced (Supplementary Fig. 12).

## The thread is highly glycosylated

Along the map of the S. acidocaldarius thread, we found dead-end protrusions that did not resemble the protein backbone or side chains. These protrusions were exclusively found at asparagine residues N56, N80, N83, N121 and N146 of each thread subunit (Fig. 1f–j). Each of these asparagine residues is part of an N-glycosylation consensus sequon (N X S/T). Since it is well known that surface exposed proteins are highly glycosylated in S. acidocaldarius[55], we concluded that these densities correspond to N-glycans. The glycan density associated with Asn146 was particularly well resolved, presumably due to its unusual parallel position to the backbone, which likely reduces its flexibility (Fig. 1j). Based on these densities, we were able to model the complete sequence of previously confirmed tribranched hexasaccharide moiety from S. acidocaldarius[56] into the filamentous structure (Fig. 1f–j). Connected to each of the five glycosylation sites at the corresponding asparagine are two N-acetylglucosamine (GlcNAc) residues, the second of which branches to bind two individual Mannose (Man), Glucose and

one 6-sulfoquinovose molecule. While glycosylation sites were previously only partially resolved in structures of crenarchaeal pili[12,44], our map allowed us to model the full N-glycan tree of S. acidocaldarius throughout the entire filament (Fig. 1e).

## Threads form cables of parallel or antiparallel filaments

We next asked how the thread filaments organise within the cables observed in our 2D classes (Supplementary Fig. 1h). Therefore, we selected these classes (containing 17,813 particles) and performed a non-helical 3D refinement. The resulting map has a resolution of 13 Å and contains 5 bundled thread filaments (Fig. 2a). The resolution was limited, likely due to variable alignment of different threads, meaning that the polarity of the filaments within the cable could not be unambiguously determined. However, by fitting the atomic model of our filament in parallel (Fig. 2b, c) and anti-parallel orientations (Fig. 2d), we found that both arrangements have equal shape complementarity. This would in principle allow the threads to interlock with one another irrespective of their polarity. In either scenario, the threads appear to interact mostly via their glycans, suggesting an important role for these post-translational modifications in cable formation.

## Threads are conserved among some Crenarchaea

To investigate how widespread threads are in archaea, we blasted Saci_0406 and found candidate proteins with a high sequence similarity in various other crenarchaea. Using SyntTax, several orthologous proteins were identified, and through use of AlphaFold2, a similar predicted structure to that of Saci_0406 was found to be encoded in the genomes of various Vulcanisaeta and Acidianus species (Fig. 3). Some homologues form a cluster of orthologs (arCOG10215) comprising members from the crenarchaeal phyla[57]. Predicting their structures using Alphafold2 suggests that these homologues are structurally almost identical to Saci_0406 (with average RMSDs of 2–3 Å; Fig. 3b–e). All five homologues follow the same beta blade head structure as the thread subunit Saci_0406. The first 7 amino acids after the N-terminal signal peptide Sec/SPI processing site are also identical, including the isopeptide Asp24 and a very conserved YYY motif (Supplementary Fig. 13). Based on the conserved Asp24, we hypothesise that the proposed isopeptide bond is conserved in the thread homologues.

Following on from this, we analysed the synteny of saci_0406 (Supplementary Fig. 14). Indeed, the genetic neighbourhood in closely related species seems to be conserved. Orthologs of saci_0406 co-occur with those of saci_0405, saci_0407 and saci_0408. RNAseq data[58] suggest that saci_0408 and saci_0407 are expressed from a different promoter than saci_0406 and saci_0405 (Supplementary Fig. 15). In some species, the genes orthologous to saci_0407 and saci_0408 are fused into one gene, whereas in others it seems that one is a noncoding region (Supplementary Fig. 14b).

## A putative cap protein

We predicted the structures of 0405, 0407 and 0408 using Alphafold2 (Supplementary Fig. 15b). While the predictions for 0407 and 0408 did not lead to conclusive insights, the Saci_0405 structure allows us to propose a role for the protein in thread assembly.

The predicted structure of Saci_0405 has a remarkably similar fold to Saci_0406 (Supplementary Fig. 16a), including the extended tail. The mean RMSD between the two structures measured only 4.58 Å (Supplementary Fig. 16b). To determine if a complex can form between Saci_0405 and Saci_0406 as is often the case between major and minor pilins, we utilised Alphafold2 to predict both proteins in a complex via the placement of the N-terminal tail of Saci_0405 into the donor strand acceptor site of Saci_0406 (Supplementary Fig. 16c). Modelling Saci_0405 into a terminal position in our thread structure revealed that Saci_0405 could complement the β1 acceptor

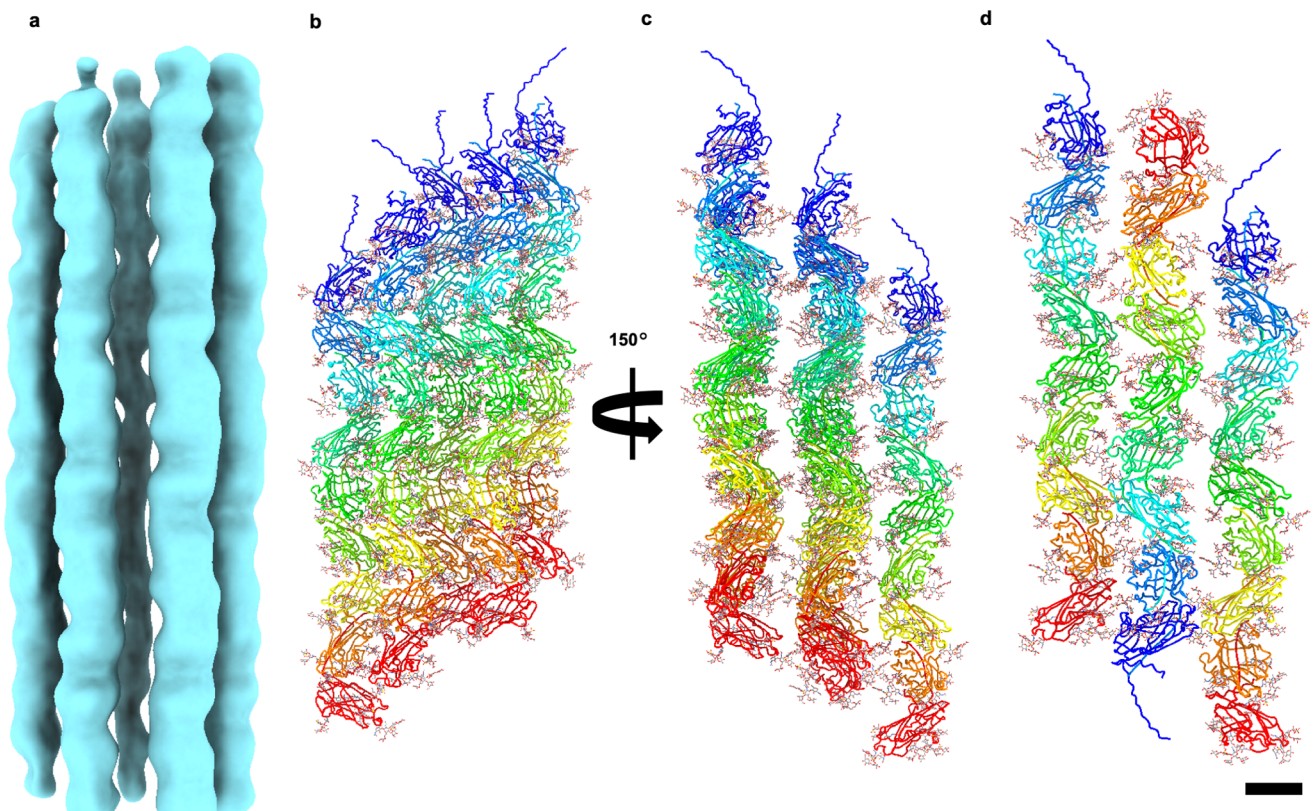

**Fig. 2 | Threads form cables of parallel or antiparallel filaments. a** CryoEM map of a thread cable at 13 Å resolution. Atomic models of the thread (in cartoon representation) can be fitted into the map in parallel (**b**, **c**) or antiparallel orientation (**d**). Shape complementarity between parallel and antiparallel threads is very similar (**c**, **d**). Glycans (sticks) mediate the interaction between the threads in the cable. A and B are in the same orientation. **c** and **d** are rotated by 150° with respect to a and b. Scale bar 40 Å.

site of a Saci_0406 n+1 subunit (Supplementary Fig. 16h, i). Interestingly, Saci_0405 lacks a donor strand acceptor site in its globular domain (Supplementary Fig. 16d), unlike Saci_0406. This intriguing finding suggests that Saci_0405 could function as a filament cap. In line with this hypothesis, Saci_0405 and Saci_0406 are very likely expressed under one promoter (Supplementary Fig. 15a), indicating co-expression and thus a functional link between the two proteins. Moreover, the RNAseq data show that *saci_0405* is expressed at lower levels and thus in lower copy numbers than *saci_0406*, in accordance with a putative function as a cap (Supplementary Fig. 15a)[58] . Cap proteins are a hallmark for bacterial filaments that assemble through DSC. All these filaments require a terminal subunit that shields the terminal tail domain and itself is not dependent on another protein to provide a completing β-strand[59].

### Threads resemble bacterial Type I pili assembled by the Chaperone Usher pathway

The overall architecture of the *S. acidocaldarius* thread filament is reminiscent of bacterial chaperone-usher *Salmonella* atypical fimbriae (Saf) pili and the recently characterised *P. gingivalis* T5P, which do not assemble via the CU pathway[35,50,60] (Supplementary Fig. 11a–d). Threads, Saf and T5P all consist of stacked subunits that are interlinked by DSC of the N-terminal β-strand. However, on the subunit level, there is little structural similarity between threads and bacterial Saf and T5P pili.

When comparing threads and bacterial T1P, there is a significant structural difference in the filaments' general architecture (Supplementary Fig. 11a, e). Whereas threads can be described as linear 40 Å wide, beads-on-a-string like filaments, T1P and P pili form hollow (likely solvent-filled) tubes with diameters of 60–70 Å[29,61]. This is reflected by

distinct helical parameters. Whereas the threads have a rise of 31.6 Å and twist of –103.2°, Type I pili have a reported rise of 7.7 Å and a twist of 115°. P pili show close similarity with values of 7.7 Å and 109.8° for rise and twist respectively[29,61]. The type V pilus has a rise of 66.7 Å with a twist of 71°.[50].

Comparing the structure of the thread subunit Saci_0406 with those of bacterial pilins reveals a striking similarity with bacterial T1P and P pilins of UPEC/UTI strains of *E. coli* (Supplementary Fig. 11a, e). Superimposing the structure of Saci_0406 with those of FimA and PapA of *E. coli* yields overall RMSD values of 9.94–11.76 Å with a highly conserved β-blade core (RMSD values around 2 Å; Supplementary Fig. 17a–d). Notably, T1P and P pili assemble via DSC, catalysed by the chaperone-usher pathway[26,28–30,61]. However, despite the structural similarity between Saci_0406 and the bacterial Type I pilins, multiple sequence alignments indicated low sequence homology (Supplementary Fig. 17e). Thus, we hypothesise that DSC and ß-sheet rich globular domains are a convergently evolved structural solution to build a remarkably stable adhesive filament.

## Discussion

Extracellular filaments are often involved in adhesive or propulsive processes and thus must be resilient to mechanical forces and adverse or changing environmental conditions. This is especially true for the threads of *S. acidocaldarius*, which inhabits environments with highly acidic pH and temperatures up to 80 °C. Threads likely gain particular tensile strength and resilience against heat and acidic pH through three structural adaptions that we have discovered: DSC, the proposed intermolecular covalent isopeptide bonds, as well as a high degree of glycosylation.

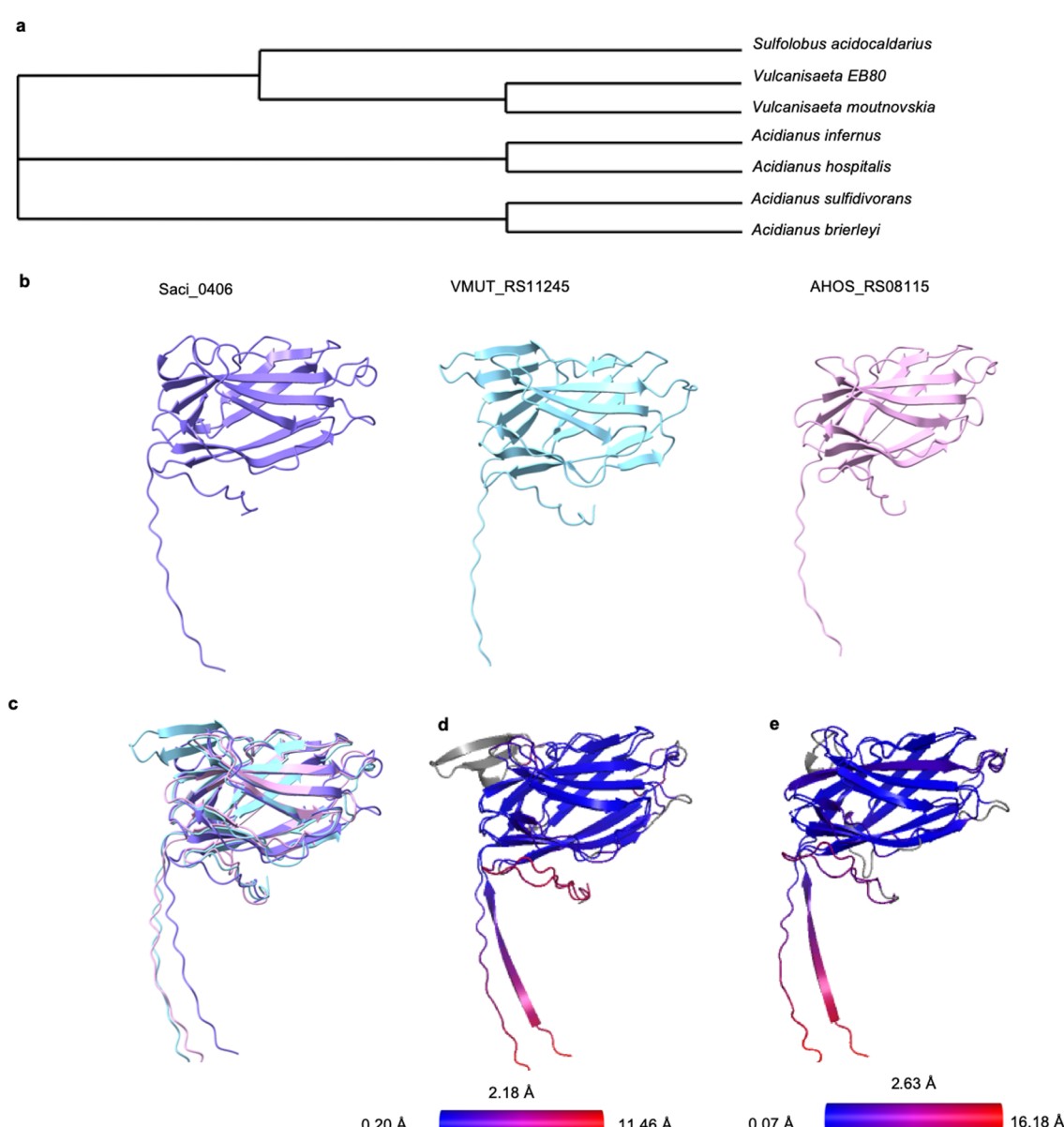

**Fig. 3 | Threads are conserved in crenarchaeal relatives. a** Phylogenetic tree of species where *Saci_0406* homologues have been identified. **b** Saci_0406 (purple) compared with Alphafold2 predictions of homologues found in *V. moutnovskia* (blue) *A. hospitalis* (pink). **c** Superimposing the structures from B shows their similarity.Pairwise RMSD values calculated between Saci_0406 and the thread homologues from *V. moutnovskia* (**d**) and *A. hospitalis* (**e**). Blue indicates low and red high values. Average value RMSD is 2.18 Å for D and 2.63 Å for E. Scale bar 20 Å.

DSC has been found previously in various types of bacterial pili, such as T1P, P-pili (CU pili), T5P[59], as well as in the very recently published structure of the archaeal bundling pili of *Pyrobaculum calidifontis*[62]. All these filaments have been implicated in adhesion or biofilm formation. In accordance with this, adhesion experiments with *S. acidocaldarius* mutants that lacked archaella, Aap and UV pili but still expressed threads, retained 25% adherence compared to WT[40]. In addition, this mutant strain was still able to form biofilms and even showed a higher density of biofilm-borne cells compared to WT[40]. We show that threads tend to align into cables of several filaments, in which parallel and antiparallel alignment between individual pili is in principle possible. This cable formation may either reinforce threads emerging from one cell or establish connections with neighbouring cells. Further experimentation will be required to elucidate the functional role of thread cables in connecting archaeal cells, as well as in the formation of biofilms.

Threads appear to be widespread in the archaeal domain, as demonstrated by our finding of *saci_0406* homologues across various crenarchaea. Multiple sequence alignments revealed a conserved SP I cleavage site in all homologues (Supplementary Fig. 13). Moreover, Alphafold2 predictions of homologues from *Acidianus hospitalis* and *Vulcanisaeta moutnovskia* reveal a very similar fold. The N-terminal ß-strand that is involved in DSC is similarly conserved, indicating that it is a shared trait in crenarchaeal threads.

Threads appear to be covalently interlinked by a non-canonical isopeptide bond between Asn57 of subunit n and the N-terminal Asp24 of the subunit $n + 2$. Strikingly, Asp24 is conserved throughout all homologues identified. Similar isopeptide bonds have been shown to provide additional stability in filaments of some Gram-positive bacteria, but these usually occur between Lys and Asn/Asp residues[54,63,64].

In SDP pili of Gram-positive species *Actinomyces oris*, an isopeptide linkage occurs between Thr499 and Lys182[65], and in *S. pyogenes*,

the C-terminal Thr311 is covalently bound to Lys161[51]. In both cases, the isopeptide bond is established between directly adjacent ($n$ and $n + 1$) subunits. In contrast, the proposed isopeptide bond in the thread is established between the subunits $n$ and $n + 2$.

In bacteria, the formation of the isopeptide bond is catalysed by a sortase. This enzyme recognises a specific motif near the C-terminus on newly incorporated subunits and cleaves the terminus that establishes the isopeptide with an upstream subunit[65]. A well-studied archaeal sortase is the archaosortase ArtA from the euryarchaeon *Haloferax volcanii*, which is essential for anchoring cell surface ArtA substrate proteins to the cell membrane. Despite similar catalytic residues, no further homology is found between ArtA and bacterial sortase enzymes[66]. As sortases have so far not been found in crenarchaea, it is likely that either a yet to be identified transpeptidase aids the formation of the isopeptide bonds, or that they form spontaneously. Indeed, spontaneous isopeptide bond formation has been proposed to occur in Gram positive bacteria such as *S. pyogenes*[63], and can be used in the engineering of novel site-specific antibody-drug conjugates[67].

As no isopeptide bonds were reported in the recently solved archaeal bundle pili of *P. calidifontis*[62], they may be a unique feature of threads. In addition, compared to the archaeal bundle pili, Saci_0406 does not contain a PFAM domain, which appears to be confined to Caldarchaeales and an uncharacterised lineage[62]. These differences suggest that threads form a separate class of archaeal cell surface filaments.

Each thread subunit is glycosylated at five asparagine residues. Based on our cryoEM map, we were able to build the complete structure of the N-glycan of *S. acidcaldarius*, which was previously sequenced via mass spectrometry[56]. In archaea, glycosylation is highly diverse, with a large variety of possible sugar moieties that may have additional modifications[55].

A high degree of glycosylation is a common feature of surface exposed archaeal proteins. For *S. acidocaldarius*, glycosylation has been shown to be crucial for motility, proper S-layer function, as well as species-specific recognition[68,69]. In addition, a high degree of glycosylation is likely an important adaptation to strongly acidic environments, and contributes to the remarkable stability of archaeal filaments, as shown for the Aap of *S. solfataricus* and *S. islandicus*[12,44]. Within the thread cables, the glycans may form molecular bridges between the individual thread strands that stabilise the superstructure via intermolecular hydrogen bonds. This highlights the importance of glycans in filament-mediated adhesion.

The structure of the thread suggests key steps in its biogenesis, as shown in Fig. 4. Bioinformatics indicates that the thread subunit Saci_0406 is secreted via the Sec pathway. In the next step, the archaeal signal peptidase I will recognise the signal peptide and cleave the N-terminus of the pre-protein. This results in the mature Saci_0406 protein with an N-terminal Aspartate and an archaea-specific triple Y motif after the cleavage site[70]. Saci_0406 will then be glycosylated by AglB, the oligosacharyltransferase which transfers the N-glycan to the specific aspartic acids within the N-glycosylation sequon[71].

The presence of DSC and the structural similarity with bacterial T1P poses the intriguing question whether the thread filament is assembled via a process akin the bacterial Chaperone Usher pathway. Here, pre-pilins are initially bound by the chaperone (FimC in Type I and PapD in P pili of *E. coli*) at the nascent N-terminus. The donor strand of the chaperone stabilises the pilin subunit until it is integrated into the growing pilus filament at the usher pore[59,72–74].

In contrast, in T5P, precursors of the subunit proteins are lipidated at a cysteine in a lipobox region. The precursors are then guided to the secondary membrane, likely by the Lol pathway, and after subsequent cleavage by an arginine/lysine specific protease the filament is polymerised[38]. In the recently published structure of the Gram-positive *Bacillus subtilis* TasA filament, the TasA subunits tightly interact via DSC forming a thin filament. In the monomeric form TasA

stabilises itself by self-complementing the binding site for the donor strand. In the filament, TasA undergoes a conformational change, and the binding site is free for DSC from the adjacent subunit. The filament polymerisation is initialised by an accessory protein TapA, which can bind to the strand accepting groove of TasA, so TasA can initiate binding to the next subunit[75]. Structural prediction of the archaeal bundle pili subunit revealed a similar mechanism[62] and indeed, we also observed auto complementation in some of our Alphafold predictions for Saci_0406 (Supplementary Fig. 16f, g).

Analysis of the gene cluster surrounding Saci_0406 did not reveal proteins that could definitively be assigned to either a chaperone or an usher protein. In line with this, *S. acidocaldarius*, and in fact most known archaea, do not possess an outer membrane, suggesting that an usher protein may not be necessary. Instead, many archaea, including *S. acidocaldarius* are encased by a proteinaceous and highly porous cell wall, called the S-layer[76]. The S-layer of *S. acidocaldarius* contains triangular and hexagonal pores with reported diameters of ~45 Å or ~80 Å, respectively[77]. With a diameter of 40 Å, threads would comfortably fit through the 80 Å hexagonal pore, which may thus serve as a conduit for the thread at the cell surface. In line with this, S-layers have previously been shown to be involved in the anchoring of archaella[78].

Saci_0405 has a fold similar to Saci_0406 with a N-terminal ß-strand but no donor strand acceptor site. This means that Saci_0405 would be able to provide its ß-strand for a downstream Saci_0406 molecule without needing a complementing ß-strand itself. Whether Saci_0405 forms a cell-proximal or cell distal cap, depends on the polarity of the thread filament, which is so-far unknown (Fig. 4). If the Saci_0406 tails point away from the cell (as it is the case for bacterial T1P, P-pili and T5P), Saci_0405 would form a cell proximal cap. In this case, Saci_0405 would terminate the thread assembly (Fig. 4, hypothesis 1). In the alternative scenario, the Saci_0406 tails point toward the cell (Fig. 4, hypothesis 2). Here, Saci_0405 acts as a nucleating subunit that initiates thread assembly and moves outward, as new Saci_0406 subunits are added to the bottom of the growing filament. In this model, Saci_0405 would play a similar role to TapA, which helps in nucleating the TasA filament formed by *B. subtilis*. A similar role has been suggested for AbpA in the recently characterised archaeal bundle pili[62,75]. Interestingly, unlike Saci_0406, the putative cap protein Saci_0405 does not have a predicted Sec signal sequence (Supplementary Fig. 9e). However, DeepTMHMM[48] predicts an extracellular localisation (Supplementary Fig. 9c). This suggests that Saci_0405 may be exported into the pseudo-periplasm via yet-to-be identified mechanism, perhaps through a membrane conduit in the thread assembly machinery.

Taken together, we propose that archaeal threads comprise a new class of archaeal biofilm-forming protein filaments that have arisen independently from similar bacterial pili. The threads are assembled by a yet unknown pathway, in which donor strand complementation likely provides the driving force . To shed more light on the biogenesis of the threads, identification and characterisation of the assembly machinery will be paramount.

## Methods

### Cell growth and threads isolation

*S. acidocaldarius* MW158 (*ΔupsEΔarlJ*) or MW114 (*ΔpibD*) were inoculated from cryostock in 6 × 5 mL basal Brock (pH 3) supplemented with 0.1% NZ-amine, 0.2% dextrin and 10 μg/mL uracil. The cells were grown for two days at 75 °C with light agitation. 5 mL of preculture was inoculated per litre main culture. The cells were grown for 48 h until they reached an $OD_{600nm} = 1$. Afterwards, cells were harvested using $5000 \times g$ for 25 min at 4 °C. The cell pellets from 2 litre main culture were resuspended in 20 mL Basal Brock (pH 3) without $FeCl_3$. To shear the threads, either shearing was done as described in Henche et al.[79] or a peristaltic pump (Gilson Minipuls) was used which was connected to syringe needle with 1.10 mm in diameter and 40–50 mm in length (Braun GmbH). The cells were homogenised at 25 rpm for 1h. The

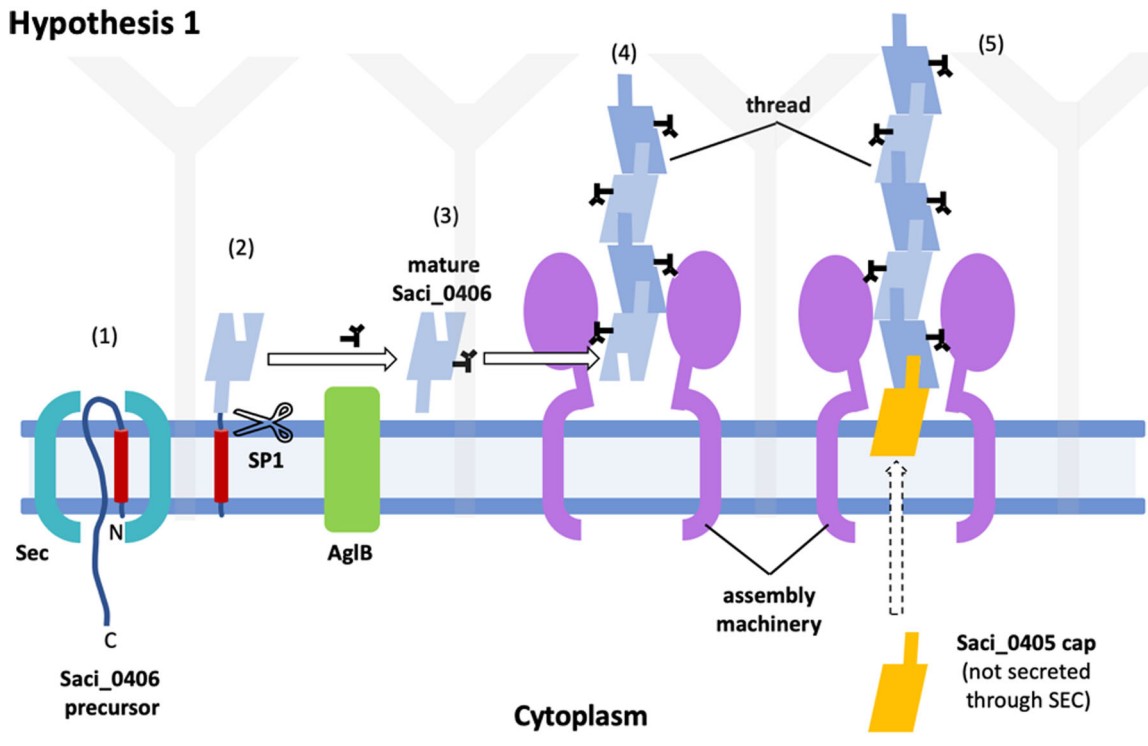

**Fig. 4 | Hypothetical models for thread assembly.** Saci_0406 precursor proteins are initially secreted through the SEC translocon (1). Signal peptidase 1 (SP1) cleaves the N-terminal signal peptide (2). The protein is glycosylated by AglB (3; glycans shown as black sticks). Soluble, mature Saci_0406 proteins assemble into the thread via donor strand complementation (4, 5). This process may be spontaneous or catalysed by an assembly machinery. The putative cap protein Saci_0405 is not predicted to be secreted through SEC. Its location in the thread depends on the polarity of the filament, which is currently unknown. If the Saci_0406 tails point away from the cell, Saci_0405 forms a cell proximal cap that terminates thread assembly (hypothesis 1). If the Saci_0406 tails point toward the cell, Saci_0405 forms a cell distal cap, which initiates thread assembly (hypothesis 2).

syringe needle was switched to more narrow ones with 0.45 mm in diameter and 10 mm and shearing was done at 25 rpm for 1 h. Subsequently, the sheared sample was centrifuged at $12,000 \times g$ for 25 min at 4 °C. The supernatant was subjected to ultracentrifugation at $200,000 \times g$ for 90 min, at 4 °C. The resulting pellet was resuspended in 500 µL Basal Brock without $FeCl_3$ and layered on 4.5 mL $CsCl_2$ (0.5 g/mL) and subjected to density gradient centrifugation at $250,000 \times g$ for 16 h at 4 °C. A white band in the upper third of the tube

was detected and collected. This was diluted to 5 mL in Basal Brock without $FeCl_3$ and pelleted at 250,000 × g for 1h at 4 °C. The pellet was resuspended in 150 μL of citrate buffer (25 mM sodium citrate/citric acid, 150 mM NaCl, pH 3) and stored at 4 °C.

## Deletion of genes in *S. acidocaldarius*

Deletion plasmids for *S. acidocaldarius* were constructed by amplifying up- and downstream flanking region of the genes of interest (400–600 bp), using the primers listed in Supplementary Table 1. Overlap PCR was performed to join upstream and downstream fragments and the joined fragment was cloned into pSVA406 containing the *pyrEF* cassette of *S. solfataricus* resulting in plasmid listed in Supplementary Table 1. The plasmid was methylated by transformation into *E. coli* ER 1821 containing pM.EsaBC4I[80]. Transformation of plasmid into *S. acidocaldarius* and generation of deletion mutant was done as described previously[81]. Deletion was confirmed by PCR and DNA sequencing. The deletion strain is listed in Supplementary Table 1.

## Negative stain transmission electron microscopy

5 μL of cells were applied on freshly glow-discharged 300 mesh carbon-coated copper grids (Plano GmbH, Wetzlar Germany) and incubated for 30 s. The excess liquid was blotted away, and grids were stained with 2% uranyl acetate. Imaging was either performed with a Zeiss Leo 912 Omega (tungsten) (Carl Zeiss, Oberkochen, Germany) operated at 80kV, equipped with a Dual Speed 2K-On-Axis charged coupled device (CCD) Sharp-Eye (TRS Systems, Moorenweis, Germany) camera, a Hitachi HT8600 operated at 100 kV, equipped with EMSIS XAROSA CMOS camera, or a FEI T12 (Thermo Fisher Scientific, Eindhoven, The Netherlands) equipped a LaB6 operating at 120 kV and a Gatan One-view CMOS detector (Gatan, Pleasanton, USA).

## CryoEM sample preparation and data collection

A 3 µl drop of suspension containing a mixture of Aap pili and threads was applied to glow-discharged 300 mesh copper R2/2 Quantifoil grids. The grids were blotted with 597 Whatman filter paper for 5 s, using -1 blot force, in 95% relative humidity, at 21 °C and plunge-frozen in liquid ethane using a Mark IV Vitrobot (FEI). Screening of grids was done using a 120 kV FEI Technai Spirit, equipped with a Gatan OneView CMOS detector (Gatan, Pleasanton, USA). High resolution image data were collected using a FEI Titan Krios electron microscope (Thermo Fisher Scientific, Eindhoven, The Netherlands), operating at 300 kV in nanoprobe mode using parallel illumination and coma-free alignment, on a Gatan K2 Summit electron detector (Gatan, Pleasanton, USA) in counting mode at a calibrated magnification of ×134,048 (corresponding to a pixel size of 1.047 Å at the specimen level) with defocus range from −2.0 to −3.5 μm using 0.3 μm steps. The microscope and camera were controlled using the EPU software (Thermo Fisher Scientific). Images were recorded as movies at a dose rate of 0.77 e/Å$^2$ s$^{-1}$ at 40 frames s$^{-1}$, 10 s exposure, with an accumulated total dose of 42.33 e/Å2. Cryo-EM statistics are presented in Table 1.

## CryoEM image processing

Frames from 6272 movies were aligned using full-frame motion correction as part of the cryoSPARC[42] package to correct for stage drift. Patch CTF estimation was then used to estimate defocus variation. The e2helixboxer programme from EMAN2[82] was used to manually pick filaments so that 2D classes could be quickly generated in Relion[83] From here, the 2D classes were imported into cryoSPARC[42], where the Filament tracer programme picked out 2,243,141 filament fragments, comprising of both AAP filaments as well as Threads. Particle coordinates were extracted from the micrographs with a box size of 512 × 512 pixel, covering 10–11 Thread subunits within a circular mask. 5 rounds of 2D classification utilising 50 classes each separated the Aap pili from the threads leading to a total of 188,620 particles from the best classes.

## Table 1 | CryoEM data collection, refinement and validation statistics

| | Thread (EMDB-13546) (PDB 7PNB) |
|---|---|
| **Data collection and processing** | |
| Magnification | 105 kx |
| Voltage (kV) | 300 |
| Electron exposure (e–/Å$^2$) | 42.33 |
| Defocus range (µm) | −2.0 to −3.5 |
| Pixel size (Å) | 1.047 |
| Symmetry imposed | Helical |
| | Twist: −103.234° |
| | Rise: 31.649 Å |
| Initial particle images (no.) | 2,243,141 |
| Final particle images (no.) | 188,620 |
| Map resolution (Å) | 3.46 |
| FSC threshold | 0.143 |
| Map resolution range (Å) | 3.6 – 3.0 |
| **Refinement** | |
| Initial model used (PDB code) | Ab initio |
| Model resolution (Å) | 3.60 / 3.43 |
| Map sharpening *B* factor (Å$^2$) | 0 |
| Model composition | |
| Non-hydrogen atoms | 15714 |
| Protein residues | 12537 |
| Ligands | 3177 |
| *B* factors (Å$^2$) | |
| Protein | 67.5 |
| Ligand | 218.1 |
| R.m.s. deviations | |
| Bond lengths (Å) | 0.008 |
| Bond angles (°) | 1.907 |
| Validation | |
| MolProbity score | 2.16 |
| Clashscore | 8.70 |
| Poor rotamers (%) | 2.56 |
| Ramachandran plot | |
| Favoured (%) | 94.48 |
| Allowed (%) | 5.52 |
| Disallowed (%) | 0.00 |

Helix Refine was used to create a non-biased 3D volume which could then be examined to determine the helical parameters. Using Chimera[84] to visualise the filament at a resolution of 7.79 Å, a rise of approximately 90 Å was predicted in real space. This was then confirmed using the symmetry search utility in cryoSPARC[42] specifying a rise range of 80-120 Å and a twist range of −180° to +180°. A sharp peak was found with a rise of ~95 Å and a twist value of ~40° which was used to refine the non-biased helix refine job, leading to a map with resolution of 4.3 Å. Closer observation of this volume revealed that the above-mentioned parameters corresponded to not 1, but 3 repeating subunits. Symmetry search was then carried out again in cryoSPARC[42], using this 4.3 Å map, to reveal the final parameters of −103° and 31.6 Å for twist and rise respectively. From here, the iterative process of performing a Local motion correction, Local CTF Refinement and Global CTF Refinement job followed by a Helix Refine job was performed, until resolution of our map reached 3.46 Å. The resolution was estimated using a Gold-Standard Fourier shell correlation between two independently refined half sets of data using the 0.143 criterion. This final map was then denoised and postprocessed using

DeepEMhancer[85]. The local resolution calculated with Local Resolution Estimation in cryoSPARC[42], and displayed in ChimeraX[86] (Supplementary Fig. 5). The map of the thread cables was generated using homogeneous (non-helical) refinement in CryoSPARC, from selected 2D classes, containing 17,813 sections of bundled filaments.

## Model building and validation

A polyalanine model was built into the observed density using Coot[87]. Side chain identities were assigned for uniquely shaped aromatic, proline and low-density glycine residues. Glycosylation sites in *Sulfolobus acidocaldarius* are predicted to be N-glycosylation and thus such sites were labelled as Asn x Ser/Thr motifs. 'X' residues were modelled as alanines or serines. Medium sized residues pointing inside the beta barrel head structure, were modelled as aliphatic leucine, isoleucine, or valine. Short surface residues were modelled as asparagine, serine or aspartate. The resulting putative amino acid sequence was run through PSI-BLAST search[88] against the *S. acidocaladarius* genome on the NCBI BLAST server[89]. The only protein which matched the search pattern with high E-value of 5e−17 (28 % sequence identity over 62% of length) was Saci_0406. The sequence of Saci_0406 could subsequently be unambiguously modelled into our map. An unusual bridging feature proved to be an isopeptide link formed between the main chain nitrogen of the N-terminal Asp24, as predicted by SignalP-5.0 server[49] and the side chain of Asn57 from the N-2 chain in the filament. To confirm our model, we predicted the structure of Saci_0406 in AlphaFold2[90], which resulted in an almost precise match for the head domain (Supplementary Fig. 8). The remaining monomers in the filament were positioned in density by phased molecular replacement implemented in MOLREP[91] adapted for EM. Later CCP4[92] script was prepared to propagate changes in the rebuilt monomer to others in the filament. The glycan structures were modelled using Coot[87], dictionaries for isopeptide link and unusual sugars were prepared using JLIGAND[93]. The structure was refined using REFMAC5[94] in the ccpem interface.

## Structural prediction (Alphafold2)

The genome of *S. acidocaldarius* was visualised using several tools. The KEGG genome database[95] allowed us to accurately determine which proteins surrounding Saci_0406 were likely to be part of the same operon. Syntax[96] was used to search for the Saci_0406 sequence to see the corresponding genomes in other archaeal species, to search for homologous gene clusters. From these sources, we concluded that Saci_0404 – Saci_0409 were likely to within the same gene cluster. Alphafold2[90] was used to predict the structures of Saci proteins 0404-0409, using the online ColabFold[97] tool. Results were observed in Chimera[84] and ChimeraX[86] to determine if any relationships between the different protein members could be determined. PDBefold[98] as well as predictions of other archaeal proteins found using Syntax were observed as stated above. Sequence analysis was done using Clustal Omega[99] to observe sequence similarities.

## Sequence analysis

Two iterative PSI-BLAST[88] searches were done using the sequence of Saci_0406 to determine over the largest sequence cover, the largest percentage identity for foreign species. The top five differing results were then structurally compared against each other using Clustal Omega[99].

## Structure analysis and presentation

Already published filaments were assessed for structural similarity to the threads. *P. gingivalis* FimA1 and *S. enterica* Saf Pilus both show DSC, similar to the threads. The structures were visualised and compared using USFC Chimera[84]. Several *E. coli* Type I pili were also studied to highlight the similarities monomerically to the threads.

## Reporting summary

Further information on research design is available in the Nature Portfolio Reporting Summary linked to this article.

## Data availability

The atomic coordinates and electron density map data generated in this study have been deposited in the protein Data Bank database under accession code 7PNB. The processed map data used in this study are available in the EM DataResource database under accession code EMDB-13546. The *S. acidocaldarius* (DSM639) genome that was analysed in this study can be accessed via the KEGG accession code T00251 or the NCBI gene bank code CP000077[100]. The transcriptomics data analysed in this study can be accessed in the Pan Genomic Database for Genomic Elements Toxic To Bacteria under this link[58].

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

## Acknowledgements

We thank Carsten Sachse at Forschungszentrum Jülich, Germany, for his helpful suggestions regarding helical reconstruction. We acknowledge Diamond for access and support of the cryoEM facilities at the UK national electron bio-imaging centre (eBIC), funded by the Wellcome Trust, MRC and BBSRC. We would also like to thank the EM facility at the Faculty of Biology at the University of Freiburg for access to their microscopes. The TEM (Hitachi HT7800) was funded by the

DFG grant (project number 426849454) and is operated by the University of Freiburg, Faculty of Biology, as a partner unit within the Microscopy and Image Analysis Platform (MIAP) and the Life Imaging Center (LIC), Freiburg. M.G., M.M. and R.U.H. were supported by an ERC Starting under the European Union's Horizon 2020 research and innovation programme (grant agreement No 803894), awarded to BD. AN was supported funding from the UK's Biotechnology and Biological Science Research Council (grant agreement number BB/R008639/1) awarded to V.G. S.S. and S.V.A. were supported by the Collaborative Research Centre SFB1381 funded by the Deutsche Forschungsgemeinschaft (DFG, German Research Foundation)—Project-ID 403222702—SFB 1381. S.S. and S.V.A. were also funded by the Deutsche Forschungsgemeinschaft (DFG, German Research Foundation) under Germany's Excellence Strategy (CIBSS–EXC-2189–Project ID 390939984).

## Author contributions

Major contributions to (i) the concept or design of the study (S.A., B.D.) (ii) the acquisition, analysis, or interpretation of the data (M.G., M.I., S.S., R.H., M.M., C.M., P.T., A.N., B.D.); (iii) writing of the manuscript (M.G., M.I., S.S., R.H., V.G., S.A., B.D.) and provision of resources (B.D., V.G., S.A.).

## Competing interests

The authors declare no competing interests
