## [Peer Review File · Nature Communications]

Electron cryo-microscopy reveals the structure of the archaeal thread filamentReviewer #1 (Remarks to the Author):

Gaines et al. describe the identification of previously observed "threads" in the crenarchaeon, *Sulfolobus acidocaldarius* using impressive structural analyses and comparisons of these structures to other prokaryotic surface filaments. However, in addition to solving the identity of the filamentous protein, the authors make claims about the functional implications of these threads (highly resilient, adapted to the harsh environment, etc.). Considering that this work does not include any functional studies, requiring the characterization of a Saci_0406 knockout, such functional implications should be removed or significantly toned down. Moreover, the introduction is missing critical information allowing a reader to understand the rationale behind the interpretations. Finally, the authors should carefully edit the manuscripts to correct numerous grammatical errors.

Additional comments

Title: The emphasis on high resilience of the filaments is not supported by the presented data. If this has not been shown elsewhere then this should be toned down (or supported by data)

Line 36: this sentence is grammatically incorrect. Moreover, while failing to sufficiently introduce the distinct prokaryotic surface filaments and their biosynthesis machineries, the authors confuse the reader by introducing many terms in the context of the threads, such as surface structures, pili, cell surface filaments, type IV filament superfamily, type IV pili and type IV pili-like filaments and threads. It should be more clearly laid out what the authors refer to when mentioning these terms.

Line 36: The authors should describe the difference between the different types of pili – particularly type IV and type I pili.

Line 44: Please spell out *S. acidocaldarius*.

Line 49: signal peptidase not signal peptide – also, what's the difference between SPIII cleaved and N-terminally processed in this context? It's misleading, suggests there is another post-translational modification

Line 58 –The phrasing of this sentence is not quite correct, as it's not the comparison of structures that shows the glycosylation, but the performed EM. Also, the "high degree of glycosylation as a protective adaptation to the highly acidic environment" is more of a hypothesis or correlation, rather than a proven causal relationship

The introduction requires more information about chaperone-usher system as well as donor strand complementation

Line 72: replace N-glycan trees with N-glycan structure or composition

Line 78: grammar/phrasing

Line 84: "However" not necessary

Line 91: why was the triple knockout not being used here?

Line 94: the delta pibD strain should not have a mixed population – type IV pili should be completely absent

Lines 146-151: The lack of the N-terminus should have been apparent in the ab initio structure already – please rewrite

Line 201: wording "intriguing to hypothesize"

Line 205: 0406 and 0407 lack saci_

Line 207: should be "in some strains" rather than "in some species"?

Line 213: Sac_0404 was not included above

Line 260: The connection to the harsh thermoacidic environment does not make sense if the same architecture is used in E. coli

Line 279: Is there any evidence from EM of cells that those cables are really connecting different cells, i.e. individual threads in the cable coming from different cells?

Line 280: The hypothesis that threads are necessary for scaffolding and special organization of the biofilm is based on what data?

Line 306: ArtA is only required for the anchoring of ArtA substrates, not all surface proteins

Line 373: "where TapA a helps" – delete "a"

Line 803: Is the Saci_0405 proposed to be the cap ion on the cytosolic side because it lacks a signal peptide? This model would suggest that the 0406 subunits need to assemble at the tip of the thread (as indicated by that white arrow). However, that tip would be far away from the cell for long threads, that do not have intra-filamental transport capabilities.

Fig. 5: Would it be possible that Saci_0405 is the cap on the outside – i.e. the first thread subunit being incorporated, and then other subunits just being added to it from the bottom? This model would not require Saci_0405 to have an N-terminal signal peptide.

Line 820: description of D, grammar; also, lengths of size bar is missing

Line 850: What is indicated by black arrow?

Line 880: Adjusting the x-axis to reflect the length of the sequence (rather than equal x-axis length despite different amino acid numbers) would help in comparing A and B

Line 889: what is the cyan-colored box in second row of sequence?

Line 955: Please provide a better description of figure 19

Reviewer #2 (Remarks to the Author):

Summary

This article by Gaines et al., describes a novel archaeal surface appendage termed the thread. These structures are highly glycosylated, presumably to enable resistance against the harsh environments in which the organism harbouring them are found. The cryo-EM structure of this filament was determined to a global resolution of 3.45 Å, which enabled the authors to identify the thread subunit as Saci_0406 and build a high-quality model of both the protein and the N-glycan moieties present. The structure revealed a unique isopeptide bond linking the n and n+2 subunits. Interestingly, the subunits are linked via donor-strand complementation, reminiscent of the pili assembled via the Chaperone Usher (CU) pathway. The N-terminal extension of each subunit complements the groove of an adjacent subunit (n+1), while also reaching all the way to the n+2

subunit, where the isopeptide linkage is located. In addition to the structure of the thread, the authors propose that Saci_0405, encoded by a nearby gene, may serve as a capping subunit for the thread filament. Overall, I think this work represents a novel and interesting discovery of a new archaeal surface appendage and will be of interest to the field. The manuscript is well written and presented. I have a few questions, as well as some minor suggestions for the authors to consider.

Major points:

The authors note that thread filaments resemble CU pili due to the donor-strand complementation visible in the structure. This immediately raises questions regarding the assembly mechanism of the threads, which may indeed be quite different. My first question is whether the thread subunits (Saci_0405 and Saci_0406) are stable on their own, or whether they indeed would require a chaperone subunit to complement their folds prior to assembly into the filament structure, as is the case for the CU subunits. Have the authors tried to express the thread subunits on their own? Perhaps there could even be self-complementation of the subunits under some conditions?

Interestingly, the authors propose that Saci_0405 might serve as a 'cap' subunit since it lacks the acceptor groove for another subunit's N-terminal extension. However, it was not entirely clear to me, depending on the section of the manuscript, whether the authors are proposing that Saci_0405 would be located at the pilus tip (distal end) or at the bottom of the pilus (cell proximal). The way Fig. 5 is drawn, Saci_0405 would be the termination subunit, analogous to PapH (P pili) and FimI (type I pili). This would leave the N-terminal extension of the tip-most Saci_0406 subunit free? How would this end of the pilus structure be satisfied? Is there a putative tip subunit present in the operon that could occupy this position? Are you confident about the filament orientation and thus about Saci_0405's relative position in the filament? Further supporting evidence around this point would significantly strengthen the manuscript.

The putative assembly mechanism in Fig. 5 is also quite distinct to that of the CU pili, given that Saci_0405 is thought not to be secreted (lacks a signal peptide), and given that the subunits are proposed to be added to the distal end of the pilus. I find it quite difficult to imagine that long filaments, like the ones visible in Supp. Fig. 1, could be assembled in such a fashion. What happens in cells lacking Saci_0405? Do such cells simply lack threads? Or, do the cells produce unusually long threads? The latter would be analogous to the phenotype of CU pili where the termination subunit is absent. This could be assessed by NS-EM.

The presence of the isopeptide bond would be quite unique to these threads. Have the authors used mass spectrometry to independently verify the presence of this linkage? Based on Fig. 1 C, it was not clear to me how well resolved the map is around the linkage. Further evidence would strengthen this finding.

Minor points

When referring to type I pili, P pili (not type P pili) or Saf pili (throughout the manuscript), it may be helpful to remind readers that these are all CU pili. Additionally, in the Abstract, it may be better and more correct to say that the donor strand complementation is reminiscent of CU pili (rather than just type I pili).

Line 26: This sentence sounds as though the filament being glycosylated, interconnected via donor-strand complementation and the presence of isopeptide bonds are all reminiscent of type 1 (CU) pili. However, the only thing that is shared amongst them is the donor-strand complementation.

Line 37: CU pili are not assembled by membrane-spanning multi-protein complexes, but rather, a single OM-embedded outer membrane protein termed usher. Assembly can of course only take place when chaperone-subunit complexes are recruited to the periplasmic domains of the usher, but neither the chaperone nor the subunit are

membrane-spanning. I would suggest rephrasing this sentence for clarity.

Line 70: Pili assembled by the CU pathway do not have isopeptide bonds. Rephrase for clarity.

Line 193: Does the NCBI protein blast server truly allow for a structural homology search? If a different tool was used here, please specify.

Line 199: Typo – “after after”

Line 204: Spelling? Syntheny/synteny?

Line 205: Please clarify if you wish to say that saci_0408 has its own promoter, or whether the whole operon is under the control of a single promoter.

Line 229: Consider rephrasing to “lower” copy number.

Line 245: Type I and P pili are both wider than 60 Å.

Line 251: UPEC and UTI strains are often the same.

Line 311: In Kang et al., 2007 (Ref 49), intramolecular isopeptide bonds were also proposed to be self-generated (sortase-independent).

Line 335: Typo: glycosylation

Figures:

For clarity and illustration purposes, it may be helpful to always orient the filament in the same way. For example, Fig. 1, Supp. Fig. 13 and perhaps also Fig. 5. This will also help to avoid confusion about whether Saci_0405 is proposed to be located at the tip or base of the pilus.

Fig. 1: It might be helpful to maintain the same colour scheme across the different panels, particularly D and E.

Fig. 5: It may be helpful for readers if the cytoplasm was indicated on the figure.

Supp. Fig. 1: In panels A-C, are the white arrow heads indicating a single thread or a bundle of threads? They appear to be too thick to only represent a single thread? Also, please indicate the scale bar on panel D. In this figure, as well as Supp. Fig. 4, I found the “white” and “black” arrow heads slightly confusing due to their borders.

Supp. Fig. 11: What is the teal box in panel C (following NNPIQ)? Is this just an image artifact?

Supp. Fig. 13: In panel B, possible typo with $\beta 1$ of $n+1$ ($n-1$?) and $\beta 1$ of $n+2$ ($n-2$?). I am not sure if this is a typo or if I read it wrongly?

Supp. Fig. 14: Perhaps you could consider indicating the glycosylated N residues? Are they conserved?

Supp. Fig. 16: I found the red, blue and black boxes difficult to see on the figure.

Supp. Fig. 18: I found it somewhat confusing to see the filament in this orientation and the ‘cap’ subunit drawn at the top of the page. Perhaps this is just me, but I got confused as to whether the authors intended to say that Saci_0405 is located at the tip or the base of the pilus. Panel E, does Saci_0405 also contain the conserved N residues?

Reviewer #3 (Remarks to the Author):

In this manuscript, Gaines et al. presented a novel structure of archaeal pilus, which reveals an interesting organization via donor strand complementation. The determined helical structure seems to be correct (not always the case) and the protein identification process is reasonable. Overall, I think this manuscript is a good candidate for Nature Communications after addressing a few issues below.

- 1) The conclusion of "isopeptide bonds". Throughout the manuscript, the only supporting evidence is the cryo-EM density shown in Fig. 1C, which is not convincing to me at the current resolution. These two residues may form a hydrogen bond instead or simply happen to be in close vicinity. In order to claim a covalent bond, the authors need to provide additional experimental data, such as mass spec result (I understand it may not be possible for this particular sample).**
- 2) The main text of the manuscript contains too many technical details and 'intermediate' results. For example, in the section of "Identifying the thread subunit protein", many of the details can go to the Method section. And the authors included too many supplemental figures in my opinion. I suggest max 10 Sup. figures instead of 22 (obviously some of them can be combined).**
- 3) In Sup. Fig.4C, "2D classification showing the tendency of threads to line up in parallel to form cables." Are those threads lined up in parallel or anti-parallel fashion? Can the authors attempt to obtain 3D reconstruction of the bundle (similar to what is shown in Fig. 3 of reference 46)?**
- 4) Can the authors provide a main figure showing the match of high-resolution side chain densities and the atomic model at a few locations?**
- 5) Related to point 4, are there homologous proteins (presumably with very similar fold) to Saci_0406 in the genome of this strain? If yes, it is critical to use the side chain densities to distinguish between these candidates.**
- 6) The current Sup Figs. 1 and 4 can be combined and serve as main Fig. 1. The current main Figs. 1 and 2 can be combined into one main figure. The current main Fig. 4 may be moved to the supplement.**
- 7) Typo on line 433, "microscop" should be "microscopy"**

Reviewer 1 - General comments:

Gaines et al. describe the identification of previously observed “threads” in the crenarchaeon, *Sulfolobus acidocaldarius* using impressive structural analyses and comparisons of these structures to other prokaryotic surface filaments. However, in addition to solving the identity of the filamentous protein, the authors make claims about the functional implications of these threads (highly resilient, adapted to the harsh environment, etc.). Considering that this work does not include any functional studies, requiring the characterization of a Saci_0406 knockout, such functional implications should be removed or significantly toned down. Moreover, the introduction is missing critical information allowing a reader to understand the rationale behind the interpretations. Finally, the authors should carefully edit the manuscripts to correct numerous grammatical errors.

Reviewer 1 - Additional comments

Comment

Title: The emphasis on high resilience of the filaments is not supported by the presented data. If this has not been shown elsewhere then this should be toned down (or supported by data)

Authors' response

We thank the reviewer for their suggestion and changed the title to avoid inferring resilience. It now says: “CryoEM reveals the structure of the archaeal thread filament”

Comments

Line 36: this sentence is grammatically incorrect.

Moreover, while failing to sufficiently introduce the distinct prokaryotic surface filaments and their biosynthesis machineries, the authors confuse the reader by introducing many terms in the context of the threads, such as surface structures, pili, cell surface filaments, type IV filament superfamily, type IV pili and type IV pili-like filaments and threads. It should be more clearly laid out what the authors refer to when mentioning these terms.

Line 36: The authors should describe the difference between the different types of pili – particularly type IV and type I pili.

Authors' combined response

The introduction has been rewritten to clearly introduce adhesive filaments, with special focus type IV and type I pili, as well as their biosynthesis pathways. Grammatical errors have been removed.

Comment

Line 44: Please spell out *S. acidocaldarius*.

Author's response

Done

Comment

Line 49: signal peptidase not signal peptide – also, what's the difference between SPIII cleaved and N-terminally processed in this context? It's misleading, suggests there is another post-translational modification

Author's response

We addressed the typo and the clarity issue in the reworked introduction.

Comment

Line 58 –The phrasing of this sentence is not quite correct, as it's not the comparison of structures that shows the glycosylation, but the performed EM. Also, the "high degree of glycosylation as a protective adaption to the highly acidic environment" is more of a hypothesis or correlation, rather than a proven causal relationship

Author's response

We agree with the reviewer's point and have removed this sentence from the reworked introduction.

Comment

The introduction requires more information about chaperone-usher system as well as donor strand complementation

Author's response

The introduction has been rewritten to include more information on chaperone-usher system as well as donor strand complementation.

Comment

Line 72: replace N-glycan trees with N-glycan structure or composition

Author's response

Done

Comment

Line 78: grammar/phrasing

Author's response

We improved the grammar of this sentence.

Comment

Line 84: “However” not necessary

Author’s response

We reworded the sentence as follows: “Although threads were already observed in the early 2000’s (23) in *S. acidocaldarius* cells, their structure and assembly mechanism has been elusive.”

Comment

Line 91: why was the triple knockout not being used here?

Author’s response

We aimed to solve the structure of threads, alongside that of AAP by imaging the same sample. We were able to solve the structure of both filaments and would now like to publish them in two separate papers. The paper on the AAP structure is currently in preparation.

Comment

Line 94: the delta pibD strain should not have a mixed population – type IV pili should be completely absent

Author’s response

To clarify; we analysed both samples. However, the sample we took forward for cryoEM was the mixed population of AAP and Threads isolated from strain MW158. For simplicity, we removed the other sample in the narrative.

Comment

Lines 146-151: The lack of the N-terminus should have been apparent in the ab initio structure already – please rewrite

Author’s response

We have made this clearer by rewriting the paragraph as follows: “Using the sequence of Saci_0406, we were able to build an atomic model for the thread unambiguously (Figure 1 b-e), excluding the N-terminus (Met1 – Ala23). We then predicted the structure of this protein using AlphaFold2. The resulting prediction closely matched our ab initio structure (Supplementary Figure 9), indicating that Saci_0406 indeed forms the subunit of the thread filaments. The AlphaFold model suggested that Met1 – Ala23 of Saci_0406 form an α -helical signal peptide, which was confirmed by the DeepTMHMM 48 server (Supplementary Figure 10 a, b). DeepTMHMM48 also predicted an extracellular localisation for Saci_0405 (Supplementary Figure 10 c). Analysing the sequence of Saci_0406 with the SignalP5 server predicted a signal peptidase I processing site, cleaving after the N-terminal α -helix between the position Ala23-Asp24 (AA sequence: ...LVA/DV...) (Supplementary Figure 10 d) 49 and explaining the absence of Met1 – Ala23 in our structure.”

Comment

Line 201: wording “intriguing to hypothesize”

Author’s response

We have rewritten this sentence as follows: “Based on the conserved Asp24, we hypothesise that the proposed isopeptide bond is conserved in the thread homologues.”

Comment

Line 205: 0406 and 0407 lack saci_

Author’s response

Corrected

Comment

Line 207: should be “in some strains” rather than “in some species”?

Author’s response

Corrected

Comment

Line 213: Sac_0404 was not included above

Author’s response

Saci_0404 was included in our initial analysis but it became clear that the gene is disrupted by a stop codon and thus not functional in *S. acidocaldarius*. We therefore removed it from the manuscript for clarity.

Comment

Line 260: The connection to the harsh thermoacidic environment does not make sense if the same architecture is used in *E. coli*

Author’s response

We appreciate the reviewer’s argument and thus changed the sentence as follows: “Thus, we hypothesize DSC and β -sheet rich globular domain is a convergently evolved structural solution to build a remarkably stable adhesive filament.”

Comment

Line 279: Is there any evidence from EM of cells that those cables are really connecting different cells, i.e. individual threads in the cable coming from different cells?

Author’s response

We solved the structure of a cable at 13 Å resolution and presented it as new main figure 2. The resolution of the model is not sufficient to draw clear conclusions on the orientation of the threads within the cable. However, it demonstrates that in principle, threads could align

in parallel, as well as antiparallel orientation. This supports our hypothesis that these cables could either reinforce filaments originating from one cell (parallel organisation) or connect neighbouring cells (antiparallel organisation). This is now detailed in a new paragraph of the discussion. We clearly observe Threads emanating from cells in negative stain images, however, it is hard to say if these actually link up with neighboring cells or if they only happen to overlap. Thus, we have toned down our interpretation and added the following statement in the discussion: “This cable formation may either reinforce threads emerging from one cell or establish strong connections with neighbouring cells. Further experimentation will be required to elucidate the functional role of thread cables in connecting archaeal cells and the formation of biofilms.”

Comment

Line 280: The hypothesis that threads are necessary for scaffolding and special organization of the biofilm is based on what data?

Author’s response

We agree that this is only speculation. We therefore removed the sentence.

Comment

Line 306: ArtA is only required for the anchoring of ArtA substrates, not all surface proteins

Author’s response

Agreed. We changed the sentence as follows: “A well-studied archaeal sortase is the archasortase ArtA from the euryarchaeon *Haloferax volcanii*, which is essential for anchoring cell surface ArtA substrate proteins to the cell membrane..”

Comment

Line 373: “where TapA a helps” – delete “a”

Author’s response

Done.

Comments

Line 803: Is the Saci_0405 proposed to be the cap ion on the cytosolic side because it lacks a signal peptide? This model would suggest that the 0406 subunits need to assemble at the tip of the thread (as indicated by that white arrow). However, that tip would be far away from the cell for long threads, that do not have intra-filamental transport capabilities.

Fig. 5: Would it be possible that Saci_0405 is the cap on the outside – i.e. the first thread subunit being incorporated, and then other subunits just being added to it from the bottom? This model would not require Saci_0405 to have an N-terminal signal peptide.

Author's combined response

The reviewer raises very valid points regarding the putative assembly mechanism, and we agree with their assessment. The thread must indeed assemble from the cell surface. We also reassessed our model regarding the thread assembly. We acknowledge that the fact that Saci_0405 is not secreted by SEC does not preclude its secretion by another (yet unknown) pathway. In consequence, we cannot tell, if it acts as a cell-proximal or a cell-distal cap. We have therefore revised our model and included two hypotheses. In the first hypothesis, Saci_0405 is not secreted and acts as a stopper subunit that terminates thread assembly. This model suggests that the N-terminal tails of the thread subunits face away from the cell, consistent with CU pili, T5P and sortase-dependent pili. However, this model would require a distal cap to protect the free N-terminal tail. A suitable candidate does not seem to be encoded in the thread operon.

In the second model, the cap sits at the distal end of the thread. In this scenario, the cap act as the first subunit of the thread, which moves further away from the cell, as new Saci_0406 subunits are incorporated. In this model, no additional cell-distal cap would be required. However, the N-terminal tails would now face towards the cell, which would be inconsistent with any of the bacterial pili that are assembled by donor strand complementation. We have presented these hypotheses as the new figure 4 and discussed them in the discussion section.

Comment

Line 820: description of D, grammar; also, lengths of size bar is missing

Authors' response

The grammar has been corrected and scale bar lengths have been included.

Comment

Line 850: What is indicated by black arrow?

Authors' response

The black arrow indicates the meridional reflection, we now highlighted this in the figure legend.

Comment

Line 880: Adjusting the x-axis to reflect the length of the sequence (rather than equal x-axis length despite different amino acid numbers) would help in comparing A and B

Authors' response

We have adjusted the x-axes to reflect the length of the sequence

Comment

Line 889: what is the cyan-colored box in second row of sequence?

Authors' response

This was an artifact of file conversion, which should now have been fixed.

Comment

Line 955: Please provide a better description of figure 19

Authors' response

An improved figure legend has been provided.

Reviewer 2 – General comments

This article by Gaines et al., describes a novel archaeal surface appendage termed the thread. These structures are highly glycosylated, presumably to enable resistance against the harsh environments in which the organism harbouring them are found. The cryo-EM structure of this filament was determined to a global resolution of 3.45 Å, which enabled the authors to identify the thread subunit as Saci_0406 and build a high-quality model of both the protein and the N-glycan moieties present. The structure revealed a unique isopeptide bond linking the n and n+2 subunits. Interestingly, the subunits are linked via donor-strand complementation, reminiscent of the pili assembled via the Chaperone Usher (CU) pathway. The N-terminal extension of each subunit complements the groove of an adjacent subunit (n+1), while also reaching all the way to the n+2 subunit, where the isopeptide linkage is located. In addition to the structure of the thread, the authors propose that Saci_0405, encoded

Reviewer 2 – Additional comments**Comment**

The authors note that thread filaments resemble CU pili due to the donor-strand complementation visible in the structure. This immediately raises questions regarding the assembly mechanism of the threads, which may indeed be quite different. My first question is whether the thread subunits (Saci_0405 and Saci_0406) are stable on their own, or whether they indeed would require a chaperone subunit to complement their folds prior to assembly into the filament structure, as is the case for the CU subunits. Have the authors tried to express the thread subunits on their own? Perhaps there could even be self-complementation of the subunits under some conditions?

Authors' response

Expressing this protein will be a very challenging undertaking, which we will address in the future but is out of scope for this manuscript. However, we tested if self-complementation is predicted by AlphaFold2 and indeed this phenomenon occurs in some of the solutions. We have included this as supplementary figure 17 f and g and discussed it in the manuscript.

Comment

Interestingly, the authors propose that Saci_0405 might serve as a 'cap' subunit since it lacks the acceptor groove for another subunit's N-terminal extension. However, it was not entirely clear to me, depending on the section of the manuscript, whether the authors are proposing that Saci_0405 would be located at the pilus tip (distal end) or at the bottom of the pilus (cell proximal). The way Fig. 5 is drawn, Saci_0405 would be the termination subunit, analogous to PapH (P pili) and FimI (type I pili). This would leave the N-terminal extension of the tip-most Saci_0406 subunit free? How would this end of the pilus structure be satisfied? Is there a putative tip subunit present in the operon that could occupy this position? Are you confident about the filament orientation and thus about Saci_0405's relative position in the filament? Further supporting evidence around this point would significantly strengthen the manuscript.

Authors' response

The reviewer raises valid points regarding the polarity of the thread and the position of the putative cap subunit. We revisited all evidence and acknowledged that we currently must consider two models; one where the Saci_0406 tails face away from the cell and Saci_0405 forms a cell-proximal cap, and another where the tails face towards the cell and Saci_0405 acts as a cell-distal cap. Both models are now shown in the new main figure 4 and have been discussed in the manuscript. See response to reviewer 1 for details.

Comment

The putative assembly mechanism in Fig. 5 is also quite distinct to that of the CU pili, given that Saci_0405 is thought not to be secreted (lacks a signal peptide), and given that the subunits are proposed to be added to the distal end of the pilus. I find it quite difficult to imagine that long filaments, like the ones visible in Supp. Fig. 1, could be assembled in such a fashion. What happens in cells lacking Saci_0405? Do such cells simply lack threads? Or, do the cells produce unusually long threads? The latter would be analogous to the phenotype of CU pili where the termination subunit is absent. This could be assessed by NS-EM.

Authors' response

We fully agree with the points raised by the reviewer. The threads must indeed be assembled at the cell surface and not at their distal end. This is now reflected by our revised assembly models in Fig. 4. In addition, we acknowledge that Saci_0405 may still be secreted by a SEC independent pathway. This means that Saci could in theory also act as a distal cap (now discussed in the manuscript). We attempted single knockouts of *saci_0405*, as well as *saci_0406*, which did not yield any colonies. This suggests that each protein alone may be toxic to the cell. However, we were able to produce a double knockout of *saci_0405* and *saci_0406* and found that this mutant does not possess any threads (new supplementary figure 1 f), which further confirms that Saci_0406 forms the thread subunit.

Comment

The presence of the isopeptide bond would be quite unique to these threads. Have the authors used mass spectrometry to independently verify the presence of this linkage? Based on Fig. 1 C, it was not clear to me how well resolved the map is around the linkage. Further evidence would strengthen this

finding.

Authors' response

To provide further evidence for the isopeptide linkage, we prepared improved figures Fig 1C, Supplementary figure 13, as well as a supplementary movie (supplementary movie 1) showing the density and fitted model in detail. As can be seen in the model, the density between Asn24 and Asp57 is particularly pronounced, similar to that of the backbone. In addition, Asn24 and Asp57 are in the positions expected for the isopeptide bond. We attempted mass spectrometry of isolated threads. However, trypsin digestion did not cleave the filaments. While we are confident that we have determined an isopeptide bond, we have toned our statements down, now referring to it as "proposed isopeptide bond".

Minor points

Comment

When referring to type I pili, P pili (not type P pili) or Saf pili (throughout the manuscript), it may be helpful to remind readers that these are all CU pili. Additionally, in the Abstract, it may be better and more correct to say that the donor strand complementation is reminiscent of CU pili (rather than just type I pili).

Authors' response

We changed "type P pili" to "P pili" throughout the manuscript and reminded the reader that Saf are CU pili, while T5P are not. The abstract now mentions "chaperone usher" instead of "Type-I pili"

Comment

Line 26: This sentence sounds as though the filament being glycosylated, interconnected via donor-strand complementation and the presence of isopeptide bonds are all reminiscent of type 1 (CU) pili. However, the only thing that is shared amongst them is the donor-strand complementation.

Authors' response

We addressed this issue by deleting "as well as isopeptide bonds" from the abstract

Comment

Line 37: CU pili are not assembled by membrane-spanning multi-protein complexes, but rather, a single OM-embedded outer membrane protein termed usher. Assembly can of course only take place when chaperone-subunit complexes are recruited to the periplasmic domains of the usher, but neither the chaperone nor the subunit are membrane-spanning. I would suggest rephrasing this sentence for clarity.

Authors' response

We have re-written the introduction for clarity.

Comment

Line 70: Pili assembled by the CU pathway do not have isopeptide bonds. Rephrase for clarity.

Authors' response

We have re-written the introduction for clarity.

Comment

Line 193: Does the NCBI protein blast server truly allow for a structural homology search? If a different tool was used here, please specify.

Authors' response

NCBI protein blast was only used to determine Saci_0406, the homologues were all determined by Syntax. Using the sequence similarity function in Synttax, we discovered several potential homologues to Saci_0406 in various crenarchaeal species. AlphaFold2 was then used to predict their 3D structures. Based on these data, further analysis into the structures of the genes surrounding these homologues was carried out.

Comment

Line 199: Typo – “after after”

Authors' response

Typo corrected

Comment

Line 204: Spelling? Syntheny/synteny?

Authors' response

Typo corrected.

Comment

Line 205: Please clarify if you wish to say that saci_0408 has its own promoter, or whether the whole operon is under the control of a single promoter.

Authors' response

Following on from this, we analysed the synteny of saci_0406 (Supplementary Figure 15). Indeed, the genetic neighbourhood in closely related species seems to be conserved. Orthologs of saci_0406 co-occur with those of saci_0405, saci_0407 and saci_0408. RNAseq data 59 suggest that saci_0408 and saci_0407 are expressed from a different promoter than saci_0406 and saci_0405 (Supplementary Figure 16). In some species, the genes orthologous to saci_0407 and saci_0408 are fused in one gene, whereas in others it seems that one is a noncoding region (Supplementary Figure 15 b).

Comment

Line 229: Consider rephrasing to “lower” copy number.

Authors’ response

Rephrased to say “lower copy numbers”

Comment

Line 245: Type I and P pili are both wider than 60 Å.

Authors’ response

Corrected to : “...T1P and P pili form hollow (likely solvent-filled) tubes with diameters of 60 - 70 Å”

Comment

Line 251: UPEC and UTI strains are often the same.

Authors’ response

Amended to : “...UPEC / UTI strains...”

Comment

Line 311: In Kang et al., 2007 (Ref 49), intramolecular isopeptide bonds were also proposed to be self-generated (sortase-independent).

Authors’ response

We have amended this sentence to include reference 49 (now 76):

“Indeed, spontaneous isopeptide bond formation has been proposed to occur in Gram positive bacteria such as *Streptococcus pyogenes* (72), and can be used in the engineering of novel site-specific antibody-drug conjugates (76).”

Comment

Line 335: Typo: glycosylation

Authors’ response

We have corrected this typo.

Figures:

Comment

For clarity and illustration purposes, it may be helpful to always orient the filament in the same way. For example, Fig. 1, Supp. Fig. 13 and perhaps also Fig. 5. This will also help to avoid confusion about whether Saci_0405 is proposed to be located at the tip or base of the pilus.

Authors' response

We have orientated the filament the same way throughout all figures.

Comment

Fig. 1: It might be helpful to maintain the same colour scheme across the different panels, particularly D and E.

Authors' response

We have addressed the reviewer's query by matching the colours across the panels of Fig. 1.

Comment

Fig. 5: It may be helpful for readers if the cytoplasm was indicated on the figure.

Authors' response

The cytoplasm is now indicated in the few figure 4.

Comment

Supp. Fig. 1: In panels A-C, are the white arrow heads indicating a single thread or a bundle of threads? They appear to be too thick to only represent a single thread? Also, please indicate the scale bar on panel D. In this figure, as well as Supp. Fig. 4, I found the "white" and "black" arrow heads slightly confusing due to their borders.

Authors' response

We confirm that the white arrowheads indicate individual threads. These may appear thicker due to the negative staining. We removed borders around arrowheads for clarity. Scale bars have been indicated throughout. We also combined supplementary figures 1 and 4 to reduce the number of supplementary figures.

Comment

Supp. Fig. 11: What is the teal box in panel C (following NNPIQ)? Is this just an image artifact?

Authors' response

The teal box was an image artefact that occurred during file conversion. This has now been removed.

Comment

Supp. Fig. 13: In panel B, possible typo with $\beta 1$ of $n+1$ ($n-1?$) and $\beta 1$ of $n+2$ ($n-2?$). I am not sure if this is a typo or if I read it wrongly?

Authors' response

We thank the reviewer for spotting these inconsistencies and corrected them.

Comment

Supp. Fig. 14: Perhaps you could consider indicating the glycosylated N residues? Are they conserved?

Authors' response

Putative glycosylation sites have now been highlighted.

Comment

Supp. Fig. 16: I found the red, blue and black boxes difficult to see on the figure.

Authors' response

We increased the thickness of the boxes, which should now be easier to discern.

Comment

Supp. Fig. 18: I found it somewhat confusing to see the filament in this orientation and the 'cap' subunit drawn at the top of the page. Perhaps this is just me, but I got confused as to whether the authors intended to say that Saci_0405 is located at the tip or the base of the pilus. Panel E, does Saci_0405 also contain the conserved N residues?

Authors' response

We rotated all filament the same way round for clarity in the former supplementary figure 18, which is now supplementary figure 17. Predicted glycosylation sites for Saci_0405 have been highlighted in panel E.

Reviewer 3 – General comments

In this manuscript, Gaines et al. presented a novel structure of archaeal pilus, which reveals an interesting organization via donor strand complementation. The determined helical structure seems to be correct (not always the case) and the protein identification process is reasonable. Overall, I think this manuscript is a good candidate for Nature Communications after addressing a few issues below.

Reviewer 3 – Additional comments

Comment

1) The conclusion of "isopeptide bonds". Throughout the manuscript, the only supporting evidence

is the cryo-EM density shown in Fig. 1C, which is not convincing to me at the current resolution. These two residues may form a hydrogen bond instead or simply happen to be in close vicinity. In order to claim a covalent bond, the authors need to provide additional experimental data, such as mass spec result (I understand it may not be possible for this particular sample).

Authors' response

We provided additional figures (Fig 1c and 13) and a supplementary movie 1 to provide further evidence for the isopeptide bond. As can be seen images and movie, the density between Asn24 and Asp57 is as strong as that of the backbone. The model with the isopeptide bond refines well and perfectly matches the density. The amino terminus of Asn24 and the carboxyl side chain of Asp57 are located in expected distances and positions with respect to each other. Attempts to refine the model with a hydrogen bond between the primary amine of Asp24 and side chain oxygen of Asn57 resulted in the Asp24 residue being pushed out of the density.

Although we are confident in our assignment, we toned down respective statements throughout the manuscript, now referring to it as “proposed isopeptide bond”. Moreover, under the heading Structure of the thread we now state: “This continuous density, as well as the distance and orientation of the involved residues are typical for isopeptide bonds that are found in SDP pili of Gram-positive bacteria (Supplementary Figure 13 a) (52–55).”

We attempted mass spectrometry of isolated threads. However, trypsin digestion did not cleave the filaments.

Comment

2) The main text of the manuscript contains too many technical details and ‘intermediate’ results. For example, in the section of “Identifying the thread subunit protein”, many of the details can go to the Method section. And the authors included too many supplemental figures in my opinion. I suggest max 10 Sup. figures instead of 22 (obviously some of them can be combined).

Authors' response

We feel that the section “Identifying the thread subunit protein” is critical to understanding how we determined the identity of the subunit. We would therefore prefer to keep it in the results part of the manuscript.

Comment

3) In Sup. Fig.4C, “2D classification showing the tendency of threads to line up in parallel to form cables.” Are those threads lined up in parallel or anti-parallel fashion? Can the authors attempt to obtain 3D reconstruction of the bundle (similar to what is shown in Fig. 3 of reference 46)?

Authors' response

We generated a 3D map of the cable at 13 Å resolution. We achieved this by running a 3D refinement on the 2D classes that contained bundles of threads. The resulting map has been included as main figure 2 in the manuscript. The limited resolution is likely due to the

threads not aligning perfectly within the cables. At the current resolution, it is not possible to tell, if the threads are aligned in parallel or antiparallel fashion. However, fitting the filaments either way showed that there is shape complementarity between the filaments in either orientation, indicating that both, parallel and antiparallel alignment is possible. Furthermore, our model shows that within the bundle, the filaments interact via their surface glycans, highlighting another important role for this post-translational modification. To our knowledge, this is the first example of a filamentous superstructure showing glycan-glycan interactions.

We added new paragraphs to the results and discussion to discuss these findings.

Comment

4) Can the authors provide a main figure showing the match of high-resolution side chain densities and the atomic model at a few locations?

Authors' response

We included two new subfigures (k and l) in figure 1 to demonstrate the quality of the data.

Comment

5) Related to point 4, are there homologous proteins (presumably with very similar fold) to Saci_0406 in the genome of this strain? If yes, it is critical to use the side chain densities to distinguish between these candidates.

Authors' response

There are no proteins with detectable sequence homology to Saci_0406 in the genome of this strain. Structure prediction using AlphaFold 2 suggests some structural similarity between Saci_0406 and Saci_0405. However, the Saci_0405 Glycosylation pattern is clearly distinguishable from that of Saci_0406 and does not correspond to the cryoEM density (supplementary figure 17e).

Comment

6) The current Sup Figs. 1 and 4 can be combined and serve as main Fig. 1. The current main Figs. 1 and 2 can be combined into one main figure. The current main Fig. 4 may be moved to the supplement.

Authors' response

We combined Supplementary Figures 1 and 4 and moved the original Fig 4 into the supplements.

Comment

7) Typo on line 433, "microscop" should be "microscopy"

Authors' response

The typo was corrected.

Reviewer #1 (Remarks to the Author):

The authors addressed most of my concerns. However, the following minor comments should be addressed.

1. In the rebuttal letter the authors explained why the mutant strain containing threads as well as Aap pili was used for the cryoEM analysis. While I understand that cryo EM of Aap pili was done simultaneously and will be published separately, not addressing why the mutant lacking all type IV pili wasn't used leaves the reader wondering, particularly since the mutant lacking all type IV pili was just mentioned in the preceding paragraph.

2. Have biofilm assays been carried out with the mutant lacking all type IV pili? If not, the authors should be more careful in suggesting that the threads are involved in biofilm formation (line 449) as in the reference cited (69) the biofilm-forming mutant only lacks archaeella and Aap pili. While in those studies, microscopy only identified the thin threads, UV pili might well be expressed during biofilm formation.

3. Line 168. "The fact the Δ saci_0406 mutant does not yield colonies, but the Δ saci_0405/0406 does, suggests that both proteins are essential for the thread formation, as well as growth of the cells." I do not understand the logic. If both genes are essential then deletion of both genes should also prevent growth.

Reviewer #2 (Remarks to the Author):

The authors have largely addressed my previous concerns. I do however still have a couple of minor comments that can be fixed by simply rephrasing and rewriting the abstract and introduction slightly:

I recommend that the sentence "We find that the filament is highly glycosylated and interconnected via donor strand complementation, reminiscent of bacterial type I pili" in the abstract is changed. To my knowledge type I pili are not highly glycosylated, so the only feature reminiscent of type I pili is that the subunits are connected via donor strand exchange. I also think the abstract should be generalised to 'chaperone-usher pili', as this feature is not unique to type I pili in particular.

I also think that lines 55-70 in the introduction are potentially confusing. The first sentence now introduces the terminology 'fimbriae', which may be confusing as the terms 'pili' and 'fimbriae' are often used interchangeably, yet here it sounds as though a whole new category of filament is introduced in this 3rd paragraph. Ref 23 appears to be about T4P from *Solifolobus*, yet the remainder of the paragraph goes on to talk about chaperone-usher pili and I'm unsure as to what the connection is that the authors intend to establish. Also, Ref 23 is perhaps no longer "very recent". The sentence "Fimbriae include Type I pili from Enterobacteriaceae and P pili from uropathogenic *Escherichia coli*, and enable bacteria to adhere to endothelial cell surfaces" is a little misleading. Uropathogenic *E. coli* also belong to the family Enterobacteriaceae and both P and type I pili are found in UPEC strains. Indeed, a lot of our knowledge about type I and P pili (both CU pili) has come from studying UPEC strains.

In lines 66-67, I would recommend that the CU pilus subunit structure would more accurately be described as consisting of an N-terminal extension (NTE) and an incomplete immunoglobulin-like fold (rather than B-barrel).

Reviewer #3 (Remarks to the Author):

The revised manuscript by Gaines et al. addressed most of my concerns and suggestions, although I still think the number of supplemental figures can be reduced (or combined). The newly added

Figure 2 is quite informative, and the new title is better. Overall, I think this revised version is significantly improved and is ready for publication in Nature Communications.

Reviewer #1 (Remarks to the Author):

The authors addressed most of my concerns. However, the following minor comments should be addressed.

1. In the rebuttal letter the authors explained why the mutant strain containing threads as well as Aap pili was used for the cryoEM analysis. While I understand that cryo EM of Aap pili was done simultaneously and will be published separately, not addressing why the mutant lacking all type IV pili wasn't used leaves the reader wondering, particularly since the mutant lacking all type IV pili was just mentioned in the preceding paragraph.

We have clarified this now by including the following in the results section: *"This strain was chosen for two reasons. Firstly, it allowed us to co-purify the threads alongside a reference filament with established purification parameters and genetic background. Secondly, analysing two structurally distinct filaments in one sample would allow us to compare their structures from the same micrographs in future studies."*

2. Have biofilm assays been carried out with the mutant lacking all type IV pili? If not, the authors should be more careful in suggesting that the threads are involved in biofilm formation (line 449) as in the reference cited (69) the biofilm-forming mutant only lacks archaella and Aap pili. While in those studies, microscopy only identified the thin threads, UV pili might well be expressed during biofilm formation.

In the cited reference 69, mutants lacking archaella, AAP **and UV pili** were tested. In these mutants, no other pili but threads were present and these cells still formed biofilms. To clarify this, we have amended the sentence in lines 328 – 330: *"In accordance with this, adhesion experiments with S. acidocaldarius mutants that lacked archaella, Aap and UV pili but still expressed threads, retained 25 % adherence compared to WT and still formed biofilms⁶⁹."*

3. Line 168. "The fact the Δ saci_0406 mutant does not yield colonies, but the Δ saci_0405/0406 does, suggests that both proteins are essential for the thread formation, as well as growth of the cells." I do not understand the logic. If both genes are essential then deletion of both genes should also prevent growth.

We clarified this query by rephrasing the sentence: *"The fact the Δ saci_0406 mutant does not yield colonies, but the Δ saci_0405/0406 does, suggests that expressing Saci_0405 alone may be toxic to the cells"*

Reviewer #2 (Remarks to the Author):

The authors have largely addressed my previous concerns. I do however still have a couple of minor comments that can be fixed by simply rephrasing and rewriting the abstract and introduction slightly:

I recommend that the sentence "We find that the filament is highly glycosylated and interconnected via donor strand complementation, reminiscent of bacterial type I pili" in the abstract is changed. To my knowledge type I pili are not highly glycosylated, so the only feature reminiscent of type I pili is that the subunits are connected via donor strand exchange. I also think the abstract should be generalised to 'chaperone-usher pili', as this feature is not unique to type I pili in particular.

Authors response:

We have rewritten the abstract to avoid suggesting that T1P are glycosylated. We also now refer to chaperone usher pili, instead of T1P for a more general comparison:

"Pili are ubiquitous filamentous surface extensions that play crucial roles for bacterial and archaeal cellular processes such as adhesion, biofilm formation, motility, cell-cell communication, DNA uptake and horizontal gene transfer. Here we report on the discovery and structure of the archaeal thread – a remarkably stable archaeal pilus that belongs to a so-far largely unknown class of highly glycosylated protein filaments. We find that the filament is interconnected via donor strand complementation, reminiscent of bacterial type chaperone-usher pili. Despite this striking structural similarity, archaeal threads appear to have evolved independently and are likely assembled by a markedly distinct mechanism."

I also think that lines 55-70 in the introduction are potentially confusing. The first sentence now introduces the terminology 'fimbriae', which may be confusing as the terms 'pili' and 'fimbriae' are often used interchangeably, yet here it sounds as though a whole new category of filament is introduced in this 3rd paragraph.

Done. "Fimbriae" has now been replaced with CU-pili

Ref 23 appears to be about T4P from *Solfolobus*, yet the remainder of the paragraph goes on to talk about chaperone-usher pili and I'm unsure as to what the connection is that the authors intend to establish.

Reference 23 was incorrect. We replaced it for the correct citation, which should now make more sense.

Also, Ref 23 is perhaps no longer "very recent".

We replaced "very recent" by "recent".

The sentence "Fimbriae include Type I pili from Enterobacteriaceae and P pili from uropathogenic *Escherichia coli*, and enable bacteria to adhere to endothelial cell surfaces" is a little misleading. Uropathogenic *E. coli* also belong to the family Enterobacteriaceae and both P and type I pili are found in UPEC strains. Indeed, a lot of our knowledge about type I and P pili (both CU-pili) has come from studying UPEC strains.

Done. We have rewritten the sentence as follows: "*CU pili include Type I and P pili, which are found in uropathogenic Escherichia coli, and enable those bacteria to adhere to endothelial cell surfaces*^{25,26}."

In lines 66-67, I would recommend that the CU pilus subunit structure would more accurately be described as consisting of an N-terminal extension (NTE) and an incomplete immunoglobulin-like fold (rather than B-barrel).

Done. The new sentence was rephrased as follows:

"Subunits for these filaments have a conserved structure consisting of an N-terminal (β -strand) extension (NTE) and a C-terminal globular domain with an incomplete immunoglobulin-like fold (head)."

Reviewer #3 (Remarks to the Author):

The revised manuscript by Gaines et al. addressed most of my concerns and suggestions, although I still think the number of supplemental figures can be reduced (or combined). The newly added Figure 2 is quite informative, and the new title is better. Overall, I think this revised version is significantly improved and is ready for publication in Nature Communications.

We thank the reviewer for their positive final assessment. We did not see a possibility to combine the supplementary figures further without adding a significant element of confusion and would therefore prefer to leave them as they are.